# Distinct RPA domains promote recruitment and the helicase-nuclease activities of Dna2

Ananya Acharya[1,2], Kristina Kasaciunaite[3], Martin Göse[3], Vera Kissling [2], Raphaël Guérois[4], Ralf Seidel [3] & Petr Cejka[1,2✉]

The Dna2 helicase-nuclease functions in concert with the replication protein A (RPA) in DNA double-strand break repair. Using ensemble and single-molecule biochemistry, coupled with structure modeling, we demonstrate that the stimulation of *S. cerevisiae* Dna2 by RPA is not a simple consequence of Dna2 recruitment to single-stranded DNA. The large RPA subunit Rfa1 alone can promote the Dna2 nuclease activity, and we identified mutations in a helix embedded in the N-terminal domain of Rfa1 that specifically disrupt this capacity. The same RPA mutant is instead fully functional to recruit Dna2 and promote its helicase activity. Furthermore, we found residues located on the outside of the central DNA-binding OB-fold domain Rfa1-A, which are required to promote the Dna2 motor activity. Our experiments thus unexpectedly demonstrate that different domains of Rfa1 regulate Dna2 recruitment, and its nuclease and helicase activities. Consequently, the identified separation-of-function RPA variants are compromised to stimulate Dna2 in the processing of DNA breaks. The results explain phenotypes of replication-proficient but radiation-sensitive RPA mutants and illustrate the unprecedented functional interplay of RPA and Dna2.

[1] Institute for Research in Biomedicine, Università della Svizzera italiana (USI), Faculty of Biomedical Sciences, Bellinzona 6500, Switzerland. [2] Department of Biology, Institute of Biochemistry, Eidgenössische Technische Hochschule (ETH), Zürich 8093, Switzerland. [3] Peter Debye Institute for Soft Matter Physics, Universität Leipzig, Leipzig 04103, Germany. [4] Institute for Integrative Biology of the Cell (I2BC), Commissariat à l'Energie Atomique, CNRS, Université Paris-Sud, Université Paris-Saclay, 91190 Gif-sur-Yvette, France. ✉email: petr.cejka@irb.usi.ch

Replication protein A (RPA) is a key single-stranded DNA (ssDNA) binding protein present in the nucleus of eukaryotic cells, which regulates most DNA metabolic processes such as DNA replication, repair, recombination, telomere maintenance and DNA damage signaling[1,2]. The key RPA functions are to protect ssDNA, remove secondary DNA structures, recruit and control DNA metabolic enzymes, and signal the presence of ssDNA to the DNA checkpoint machinery. While simple ssDNA binding by RPA is sufficient for some processes, many RPA functions depend on specific physical and functional interactions between RPA and its cofactors[1,2]. Depending on its partners, RPA thus regulates opposing processes such as DNA synthesis and nucleolytic degradation, as well as DNA unwinding and annealing[3].

RPA is a modular heterotrimeric protein consisting of three subunits, RPA1 (Rfa1 in the budding yeast Saccharomyces cerevisiae), RPA2/Rfa2 and RPA3/Rfa3. Each subunit contains oligonucleotide-binding (OB) domains that mediate ssDNA binding. The large Rfa1 subunit contains DNA-binding domains F, A, B and C. The Rfa2 protein contains Rfa2-D and Rfa3 possesses Rfa3-E domains. The Rfa1-AB domains exhibit the highest affinity to ssDNA, while Rfa1-C, Rfa2-D and Rfa3-E are secondary ssDNA binders, which are additionally responsible for the trimerization of the RPA complex[4–7]. The N-terminus of Rfa1 (Rfa1-F) provides only a minor, if any, contribution to DNA binding, and rather mediates specific physical interactions with RPA-binding proteins, along with the C-terminus of Rfa2[1,8,9]. Although ssDNA binding of the heterotrimeric RPA is very strong, it is highly dynamic. Due to the modular mechanism of DNA binding, RPA partners that per se exhibit much lower affinity to ssDNA can displace RPA by targeting sequentially its individual DNA-binding domains[7,10–12]. RPA binding to ssDNA is directional. Rfa1-A along with the Rfa1-F localize to the 5′-end of ssDNA, while the Rfa2-D domain binds the opposite 3′-end[4,13]. Such association of RPA with ssDNA can help enforce directionality of the linked downstream metabolic processes[13].

Dna2 is a conserved nuclease-helicase that likely functions in DNA replication during Okazaki fragment processing along with FEN1/Rad27, and in DNA double-strand break (DSB) repair by homologous recombination together with a RecQ family helicase during the initial DNA end resection step[14]. Dna2 may have an additional lesser-defined function in DNA replication stress[15]. In Okazaki fragment processing, Dna2 likely cleaves long 5′-terminated ssDNA flaps that arise upon DNA displacement synthesis by DNA polymerase δ in conjunction with Pif1 on the lagging DNA strand. Short flaps are cleaved by FEN1, but flaps that are long enough to bind RPA become refractory to FEN1 cleavage, and instead become a substrate for Dna2[16–20]. RPA was shown to stimulate the cleavage of DNA flaps, which was explained by its capacity to recruit Dna2 to the substrate[16,21,22]. A structural study with mouse DNA2 revealed that the nuclease domain of DNA2 contains a narrow tunnel, through which ssDNA needs to thread before being cleaved, indicating that RPA must be displaced from ssDNA before DNA threading and degradation can take place[22].

In DNA end resection, Dna2 functions in conjunction with a RecQ family helicase, such as Sgs1 in Saccharomyces cerevisiae or Bloom (BLM) or Werner (WRN) in human cells[23–25]. Sgs1/BLM/WRN is the lead motor that unwinds dsDNA from the broken ends. RPA then directs the unwound 5′-terminated strand to the nuclease domain of Dna2. Without RPA, Dna2 degrades ssDNA with both 5′ and 3′ polarities, but under physiological conditions when RPA is present, Dna2 only cleaves the 5′-strand in agreement with the DSB repair models[26,27]. Therefore, RPA enforces the correct polarity of DNA end resection, which is thought to be a consequence of RPA's directional binding to ssDNA. Beyond its nuclease, Dna2 also possesses a conserved helicase domain with a cryptic unwinding activity, which is within the wild type protein blocked by its own nuclease function[28,29]. In DNA end resection, the motor activity of Dna2 does not likely function to unwind dsDNA, but rather as a ssDNA translocase to speed up the movement of Dna2 along unwound ssDNA downstream of Sgs1/BLM/WRN. The enhanced velocity promotes ssDNA degradation by the Dna2 nuclease domain, in particular when RPA is present[30–32].

Both enzymatic activities of Dna2 are stimulated by RPA in a specific manner, but the mechanism by which it occurs remains poorly defined. Dna2 was shown to physically interact with the N-terminal domain of Rfa1 (Rfa1-F), with secondary binding sites located more downstream in Rfa1-AB and Rfa1-C[22,33]. The physical interaction was proposed to help recruit Dna2 to the DNA substrate[16,21], and in particular to the 5′-terminated DNA strand in agreement with the RPA's binding directionality. RPA-mediated Dna2 recruitment to DNA was thought to explain the stimulatory activity of RPA on the enzymatic activities of Dna2. We show here that RPA promotes the catalytic activities of Dna2 in addition to its recruitment function. Using mutational analysis, biochemistry, structure modeling and single-molecule biophysics, we show that the domains of RPA required for the stimulation of Dna2 recruitment, Dna2 nuclease and helicase activities are not identical. Whereas both Rfa1-F and Rfa1-C domains of Rfa1 are involved in Dna2 recruitment, only the Rfa1-F domain in conjunction with Rfa1-AB but not Rfa1-C are needed for the stimulation of the nuclease activity. We identify mutations in the N-terminal Rfa1-F domain that are nearly essential for the stimulation of the Dna2 nuclease activity, but the respective residues are not involved in Dna2 recruitment or stimulation of the Dna2 helicase. In contrast, both Rfa1-F and Rfa1-C domains are dispensable for the stimulation the Dna2 helicase, while the Rfa1-AB domains are necessary and sufficient for this activity. Using three-dimensional molecular docking, we identify residues that mediate the specific functional interaction between Dna2 and Rfa1-A, which were validated and confirmed to be required for the stimulation of the Dna2 motor. Our experiments define the multifaceted interplay of Dna2 and RPA, which is required for proper function of Dna2 in DNA end resection.

## Results

**RPA specifically promotes both helicase and nuclease activities of Dna2.** Both nuclease and helicase activities of S. cerevisiae Dna2 are known to be stimulated by RPA[14,29]. In accord, we show that cognate heterotrimeric RPA promoted the nuclease activity of wild type Dna2 (Fig. 1a–e), and the helicase activity of the nuclease-dead Dna2-E675A, unlike the non-cognate factors (Fig. 1f–h). Human mitochondrial ssDNA binding protein (mtSSB), which bears no sequence similarity to RPA, and rather resembles prokaryotic SSB, in contrast inhibited both activities of Dna2 (Fig. 1d, f–h), the same results were also obtained with other Dna2 concentrations. DNA unwinding by the Dna2 partner Sgs1 (Supplementary Fig. 1a) under the same conditions was instead promoted similarly by either mtSSB or RPA (Supplementary Fig. 1b–f). RPA activates DNA unwinding by Sgs1 and stimulates its processivity in single-molecule assays[34,35]. A proportion of this apparent stimulatory effect likely involves simple prevention of DNA reannealing, which does not require specific interactions between Sgs1 and mtSSB. In contrast, specific functional and physical interactions clearly underpin the interplay of Dna2 and RPA.

**RPA promotes Dna2 beyond recruitment.** Previously, RPA was shown to recruit Dna2 to DNA[21], and the recruitment function was thought to explain the apparent stimulation of Dna2

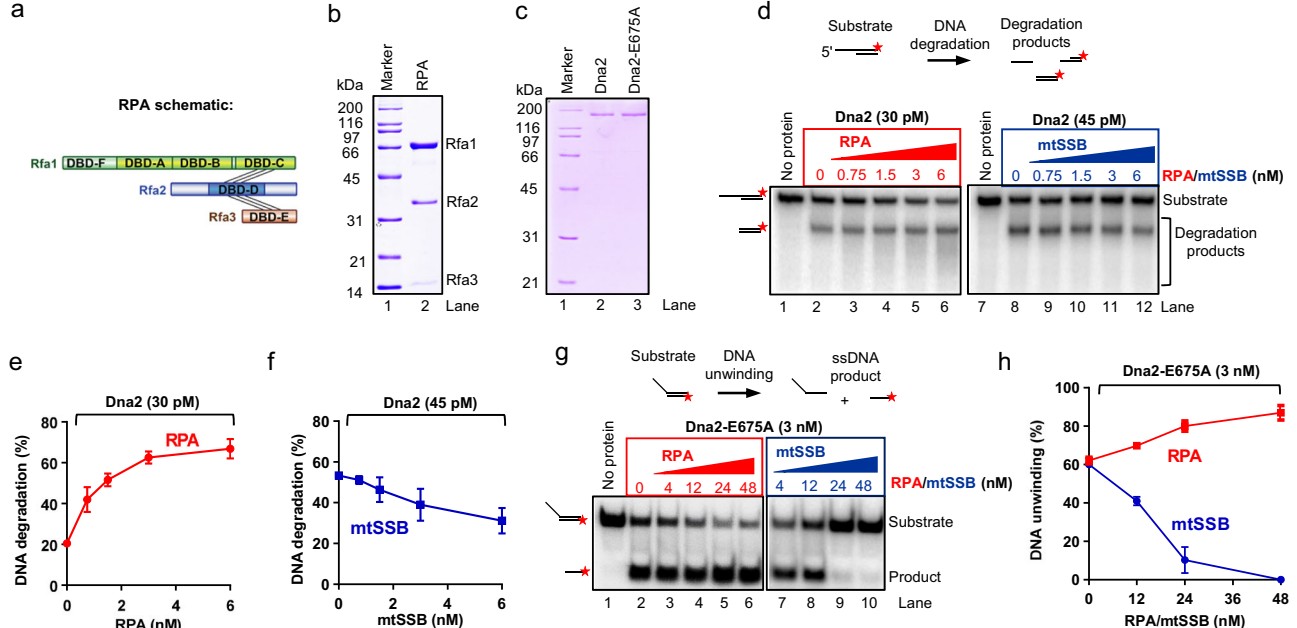

**Fig. 1 RPA specifically promotes both helicase and nuclease activities of Dna2. a** A schematic of RPA. **b** Purified wild-type RPA used in this study. **c** Recombinant *S. cerevisiae* Dna2 and Dna2-E675A (nuclease-dead) variants used in this study. **d** Yeast RPA and human mitochondrial SSB (mtSSB) were used in nuclease assays with Dna2 and 5′- overhanged DNA substrate (45 nt ssDNA, 48 bp dsDNA, 1 nM, in molecules). The red asterisk indicates the position of the radioactive label. **e, f** Quantification of nuclease assays such as shown in panel **d**. Error bars, SEM; $n = 3$. **g** Yeast RPA and mtSSB were used in helicase assays with Dna2-E675A and 5′-overhanged DNA substrate (30 nt ssDNA, 31 bp dsDNA, 1 nM, in molecules) with 50 mM KCl. **h** Quantification of assays such as shown in panel **g**. Error bars, SEM; $n = 4$.

enzymatic activities[21,22]. It was proposed that Dna2 immediately replaces RPA on the DNA substrate[21], although this conclusion was also disputed[16,22]. We observed that low concentrations of Dna2 (5–10 nM) were inefficient to bind overhanged DNA on their own (Fig. 2a, lanes 3 and 4, Fig. 2b). The same Dna2 concentrations led to a clear electrophoretic mobility shift when RPA was pre-bound to the substrate (Fig. 2a, lanes 9 and 10), showing that RPA facilitates the binding of Dna2 to DNA. In contrast, no DNA binding by Dna2 was observed when DNA was pre-bound by mtSSB (Fig. 2a, b), suggesting that mtSSB prevents the access of Dna2 to DNA. To distinguish whether Dna2 replaced RPA or whether both proteins bound DNA simultaneously, we characterized the formed complexes by mass photometry. This technique allows one to estimate the molecular weight of biomolecules and their complexes at nanomolar concentrations by evaluating the backscattered light from individual complexes in a microscope setup (Fig. 2c). RPA alone exhibited a single dominant peak in the mass spectrum at a molecular weight 115 ± 27 kDa (theoretical value 114 kDa). Low concentrations of Dna2 alone showed molecular weight of 166 ± 28 kDa (theoretical value 172 kDa), which did not change when DNA was added, indicating inefficient DNA binding (Fig. 2d and Supplementary Fig. 2a). When RPA was added to Dna2 without DNA, we observed only mass peaks corresponding to single RPA and Dna2 molecules (Fig. 2d). When DNA was added to the Dna2 and RPA sample, we observed a new species at 302 ± 67 kDa, corresponding to the combined molecular weights of Dna2, RPA, and DNA (theoretical total weight of 311 kDa). The species corresponding to Dna2 and DNA (202 kDa) was not observed. RPA-DNA species were positioned between the peaks of single RPA and Dna2s and were not easily distinguishable. Therefore, we conclude that RPA recruits Dna2 to DNA, and remains a component of the nucleoprotein complex at least as an intermediate. RPA is therefore positioned to promote Dna2 activities in principle also downstream of recruitment.

To establish whether RPA promotes Dna2 beyond recruitment, we prebound Dna2-E675A to DNA, using high Dna2 concentrations to achieve efficient binding (Fig. 2e), employing conditions to limit DNA unwinding by Dna2-E675A alone. We then activated unwinding reactions by adding ATP, or ATP at the same time with SSB or RPA (Fig. 2f). In this setup, when Dna2-E675A did not need to be recruited, RPA still promoted the helicase activity of Dna2-E675A in a specific manner, much more than SSB that acts non-specifically by preventing DNA reannealing. Therefore, RPA specifically promotes the Dna2 activities beyond facilitating recruitment to DNA, which is further supported by mutational analysis later in this study.

### The Rfa1-FAB domains stimulate the Dna2 nuclease. To define how RPA stimulates Dna2, we set out to prepare RPA point mutants and truncations. As some of these mutants exhibited reduced ssDNA binding that could have impaired the established protein purification procedure, we modified the RPA preparation protocol to include an affinity step upon adding a his-tag on the N-terminus of Rfa1 (Supplementary Fig. 3a, b). Untagged and his-tagged RPA bound ssDNA indistinguishably and both variants similarly stimulated the Dna2 helicase and nuclease activities (Supplementary Fig. 3c–f), showing that the his-tag did not impair RPA functions relevant to the interplay with Dna2.

We first tested point mutants in the N-terminal domain of Rfa1 (Rfa1-F), including I14S and K45E (Fig. 3a), which were initially identified in a screen for RPA mutations resulting in radiation sensitivity but could support growth otherwise[36]. This phenotype suggested proficiency in DNA replication, but potential defects in DNA repair processes such as recombination. Additionally, the *rfa1-I14S* mutation was later found to be synthetically lethal in a *dna2* helicase-deficient background, and was demonstrated to impair physical interaction with Dna2[33]. We also selected mutations mapping to the Rfa1-C domain, which

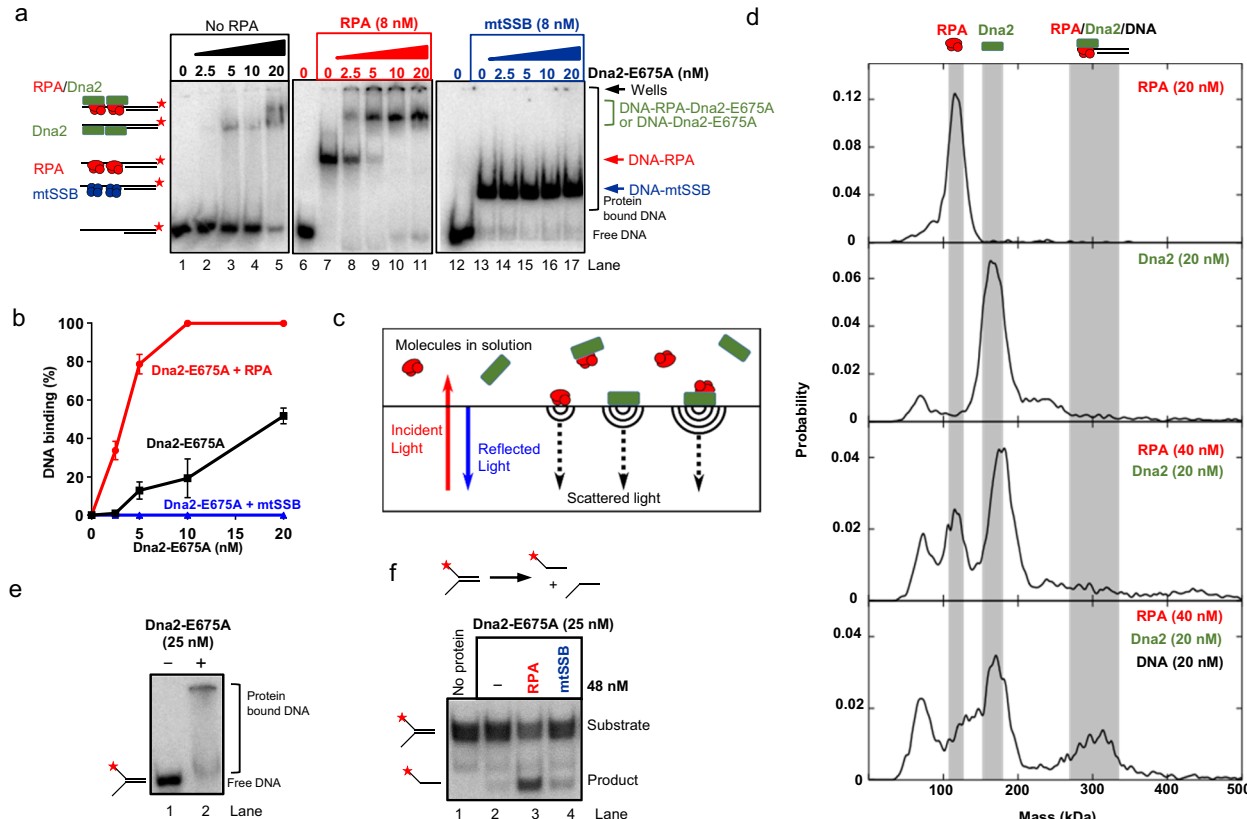

**Fig. 2 RPA stimulates Dna2 beyond recruitment to DNA. a** Representative electrophoretic mobility shift assays to monitor recruitment of Dna2-E675A to either free or RPA/mtSSB-precoated (8 nM) 5′-overhanged DNA substrate (45 nt ssDNA, 48 bp dsDNA, 1 nM, in molecules) and with 150 mM NaCl. The red asterisk indicates the position of the radioactive label. **b** Quantification of assays such as shown in panel **a**. Error bars, SEM; $n = 3$. **c** A scheme of molecular weight determination using mass photometry. Scattered light from single adsorbed biomolecular complexes interferes with light reflected at the surface. The detected single-molecule interference contrast is proportional to the molecular weight. **d** Measured molecular weight distributions of RPA and Dna2 complexes in absence and presence of a 5′-overhanged DNA substrate (25 nt ssDNA, 48 bp dsDNA, 20 nM, in molecules). For the formation of a heterotrimeric complex (panel at the bottom), RPA was added first to the DNA followed by Dna2 addition. **e** DNA binding of Dna2-E675A to forked DNA (45 nt ssDNA, 48 bp dsDNA, 0.1 nM, in molecules) and with 50 mM KCl. One out of two independent experiments is shown. **f** Unwinding of forked DNA as in panel **e** by Dna2-E675A. DNA substrate was first pre-bound by saturating concentrations of Dna2-E675A (25 nM) before adding yeast RPA or human mtSSB (both 48 nM) together with ATP and with 50 mM KCl. One out of three independent experiments is shown.

impair ssDNA binding, creating RPA-K494E and RPA-CCAA (C505A, C508A) (Fig. 3a, b, and Supplementary Fig. 3g)[36,37]. Accordingly, the RPA-CCAA mutations are lethal in yeast[37]. As anticipated, the I14S and K45E mutations did not impair ssDNA binding of the RPA heterotrimer, while the K494E and CCAA mutations reduced ssDNA binding ~3-fold, based on apparent $K_D$ (Fig. 3a, b, and Supplementary Fig. 3h). We then analyzed the ability of the RPA variants to stimulate the Dna2 nuclease. We observed that RPA-I14S, and to a lesser degree RPA-K45E, reduced but did not eliminate the stimulation of the Dna2 nuclease, while the mutations in the Rfa1-C domain did not have a notable effect (Fig. 3c). The RPA-K45E mutant was previously found to be impaired also in its interaction with the MRX complex in yeast[38]. As the N-terminal domain of Rfa1 does not significantly bind DNA, its contribution to the stimulation of the Dna2 nuclease likely reflects specific physical and functional interactions with the Dna2 nuclease in agreement with previous data[22,33].

Next, we expressed and purified the RPA subunits Rfa1, Rfa2 and Rfa3 individually, as well as various truncations of the main subunit Rfa1 (Fig. 3d and Supplementary Fig. 3i, j). As anticipated, the variants differed in affinity to ssDNA (Fig. 3d, e, and Supplementary Fig. 3k, l). We observed that Rfa1 alone was fully sufficient to catalyze the stimulation of the Dna2 nuclease,

while the Rfa2 and Rfa3 subunits possessed no detectable stimulatory activity (Fig. 3f, g). Within Rfa1, we found that the N-terminal Rfa1-F, along with Rfa1-A and Rfa1-B domains, were essential for the stimulation of the Dna2 nuclease, while the Rfa1-C domain, which provides the strongest contribution towards ssDNA binding, was entirely dispensable (Fig. 3h–j)[33]. We also noted that the Rfa1-F domain needs to be part of the same polypeptide together with Rfa1-AB, as no stimulation was observed when individual truncations were combined (Fig. 3k and Supplementary Fig. 3m, n).

**RPA-SSL mutant of the Rfa1-F domain fails to stimulate the Dna2 nuclease.** To better define the contribution of the residues around I14 of Rfa1 to the stimulation of the Dna2 nuclease, we performed molecular modeling to predict the impact of the mutation on the structure of Rfa1-F. The residue I14 is located in a conserved helix embedded in the N-terminal domain, and is thus unlikely to mediate protein-protein interaction directly (Fig. 4a–c). Rather, mutation of this residue could result in alteration of the surrounding structure including residues that are exposed to the surface, impacting thus the interplay with Dna2. We next replaced two additional conserved residues in the vicinity of I14 in the same helix, namely F11 and F15, to prepare the

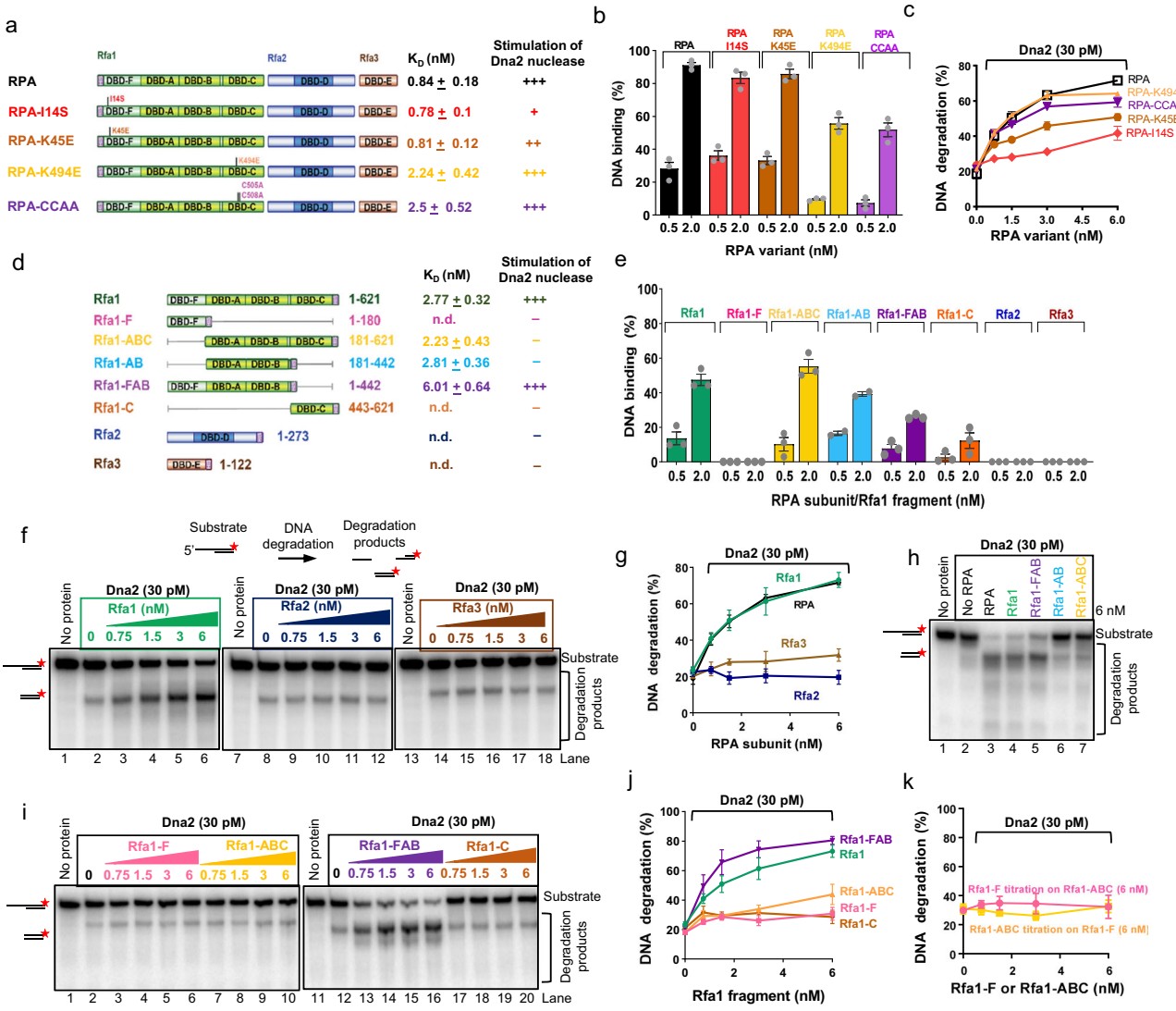

**Fig. 3 The DBD-FAB domains of Rfa1 stimulate the Dna2 nuclease. a** A scheme of wild type RPA and selected point mutants. $K_D$, concentration of the respective RPA variant resulting in 50% binding of ssDNA (93 nt, 0.1 nM, in molecules) such as shown in Supplementary Fig. 3c, h. Error, SEM; $n = 3$. The extent of stimulation of the Dna2 nuclease is indicated on the right. **b** Quantitation of ssDNA (93 nt, 0.1 nM, in molecules) binding by the RPA variants as shown in panel **a**. Error bars, SEM; $n = 3$. **c** Quantification of nuclease assays with his-tagged RPA wild type and point mutants and Dna2 using 5′-overhanged DNA (45 nt ssDNA, 48 bp dsDNA, 1 nM in molecules). Error bars, SEM; $n = 3$. **d** A scheme of RPA subunits and Rfa1 fragments. $K_D$, concentration of the respective variant resulting in 50% binding to ssDNA (93 nt, 0.1 nM, in molecules), such as shown in Supplementary Fig. 3k, l; Error, SEM; $n = 3$. The extent of stimulation of the Dna2 nuclease is indicated on the right, based on nuclease assays such as shown in panels **f** and **i**. **e** Quantitation of ssDNA (93 nt, 0.1 nM, in molecules) binding by Rfa1 fragments shown in panel **d**. Error bars, SEM; $n = 3$. **f** Representative nuclease assays showing degradation of 5′-overhanged DNA (45 nt ssDNA, 48 bp dsDNA, 1 nM in molecules) by Dna2 and its stimulation by the RPA subunits. The red asterisk indicates the position of the radioactive label. **g** Quantification of nuclease assays such as shown in panel **f**. RPA wild type is replotted as in panel **c** for reference. Error bars, SEM; $n = 3$. **h** Representative nuclease assays as in panel **f**, but with shorter 5′-overhanged DNA (19 nt ssDNA, 31 bp dsDNA, 1 nM, in molecules) and with 100 mM NaCl. **i** Representative nuclease assays with Rfa1 fragments and Dna2 using 5′-overhanged DNA (45 nt ssDNA, 45 bp dsDNA, 1 nM, in molecules). **j** Quantification of nuclease assays such as shown in panel **i**. Rfa1 is replotted as in panel **g** for reference. Error bars, SEM; $n = 3$. **k** Quantification of nuclease assays such as shown in Supplementary Fig. 3m, n. Error bars, SEM; $n = 3$.

RPA-SSL (F11S, I14S, F15L) mutant (Fig. 4d and Supplementary Fig. 4a). The triple mutations within the RPA heterotrimer were anticipated to exacerbate the effect of I14S. While the RPA-I14S mutant bound ssDNA indistinguishably from wild type RPA, the SSL mutant exhibited ~1.6-fold lower ssDNA binding affinity (Fig. 4e and Supplementary Fig. 4b). This reduction of ssDNA binding was however more subtle than that of the RPA-K494E and RPA-CCAA variants (Fig. 3b, ~3-fold reduction), which did not affect the stimulation of the nuclease. Therefore, the decrease in ssDNA binding of RPA-SSL is unlikely to be responsible for any potential effects on the stimulation of the Dna2 nuclease.

Next, we compared the capacity of wild type RPA and the RPA-I14S and RPA-SSL mutants to stimulate Dna2 to cleave 5′-overhanged DNA. The SSL variant was dramatically impaired to promote the Dna2 nuclease, much more so than RPA-I14S (Fig. 4f, g). As described above, Rfa1 alone can stimulate the Dna2 nuclease, similarly as heterotrimeric RPA (Fig. 3g). Accordingly, the I14S mutation within the Rfa1 subunit alone had a similar negative effect, comparable to the I14S mutation within the heterotrimer. Rfa1-K494E mutant was able to stimulate the Dna2 nuclease better than the Rfa1-I14S, despite its lower DNA-binding capacity (Supplementary Fig. 4c–f), in agreement with

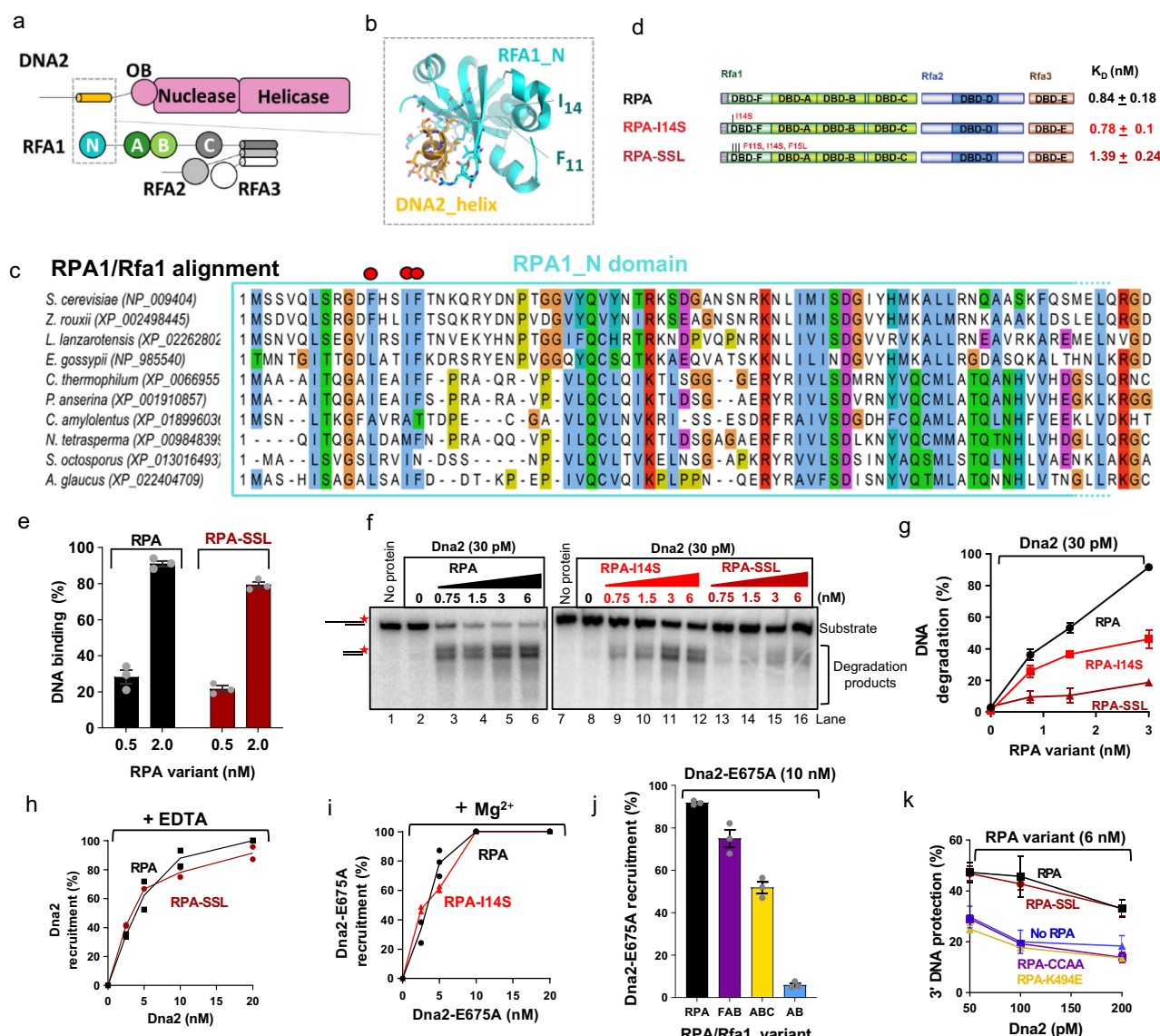

**Fig. 4 Dna2 recruitment and nuclease stimulation by RPA are genetically separable. a** A schematic representation of the domain organization in Dna2 and RPA. **b** A scheme of the interaction between the N-terminal domain of *S. cerevisiae* Rfa1 (DBD-F) and the predicted helix within the N-terminal part of Dna2 generated by comparative modeling using the structure of the orthologous human complex (PDB: 5EAY). Residues I14 and F11 mutated in this study are highlighted. **c** Multiple sequence alignment of RPA1/Rfa1 orthologs from various species focused on the N terminal region of RPA1, emphasizing the conserved positions located in the interface of the structural model of the complex between yeast Rfa1 and Dna2. **d** A schematic representation of the primary structures of RPA-SSL mutant. Wild type RPA and RPA-I14S are again shown as reference. $K_D$, concentration of the respective RPA variant resulting in 50% binding to ssDNA (93 nt, 0.1 nM, in molecules,) such as shown in Supplementary Fig. 4b; Error, SEM; $n = 3$. **e** Quantitation of ssDNA (93 nt, 0.1 nM, in molecules) binding by wild type RPA and RPA-SSL mutant in assays such as shown in Supplementary Figs. 3c and 4b. RPA is replotted as in Fig. 3b for reference. Error bars, SEM; $n = 3$. **f** Representative nuclease assays using Dna2 and 5′-overhanged DNA substrate (45 nt ssDNA, 48 bp dsDNA, 1 nM, in nucleotides) performed as in Fig. 1d, but with 100 mM NaCl. The red asterisk indicates the position of the radioactive label. **g** Quantification of nuclease assays such as shown in panel **f**. Error bars, SEM; $n = 3$. **h** Quantification of DNA binding assays such as shown in Supplementary Fig. 4g. Wild type Dna2 was bound to 5′-overhanged DNA substrate (45 nt ssDNA, 48 bp dsDNA, 1 nM, in molecules) pre-coated with either wild type RPA or RPA-SSL in the presence of 3 mM EDTA and 150 mM NaCl. Bars indicate range; $n = 2$. **i** Quantification of experiments as in panel **h**, but with Dna2-E675A in the presence of 5 mM $Mg^{2+}$ and 150 mM NaCl. RPA is replotted as in Fig. 2a for reference. Bars indicate range; $n = 2$. **j** Quantification of assays such as shown in Supplementary Fig. 4h showing Dna2-E675A binding to 5′-overhanged DNA substrate (30 nt ssDNA, 31 bp dsDNA, 1 nM, in molecules) pre-coated with RPA, Rfa1-FAB, Rfa1-ABC or Rfa1-AB fragments in the presence of 5 mM $Mg^{2+}$ and 150 mM NaCl. Error bars, SEM; $n = 3$. **k** Quantification of assays such as shown Supplementary Fig. 4i. The respective RPA variant was tested for capacity to protect 3′-overhanged DNA substrate (45 nt ssDNA, 48 bp dsDNA, 1 nM, in molecules) from Dna2 degradation with 100 mM NaCl. Error bars, SEM; $n = 4$.

results from experiments where these mutations were analyzed within the RPA trimer (Fig. 3a–c). We conclude that the integrity of the region surrounding residues F11, I14 and F15 in the N-terminal Rfa1-F domain is essential for RPA to stimulate the nuclease activity of Dna2.

**Dna2 recruitment and nuclease stimulation by RPA are genetically separable.** The stimulation of the Dna2 nuclease was previously attributed to the recruitment function of RPA[21,22]. However, we observed that the RPA-I14S and RPA-SSL mutants were not impaired in their function to recruit Dna2 to

overhanged DNA (Fig. 4h, i, and Supplementary Fig. 4g), which challenges the previous model. To understand the relationship between the stimulation of the Dna2 nuclease and recruitment to DNA in more detail, we next analyzed the capacity of the various Rfa1 domains to mediate recruitment. RPA could recruit Dna2, while Rfa1-FAB, and Rfa1-ABC were moderately impaired in this function. In contrast, Rfa1-AB could not recruit Dna2 at all, despite its strong DNA-binding capacity, demonstrating the importance of both Rfa1-F and Rfa1-C domains of Rfa1 for Dna2 recruitment (Fig. 4j Supplementary Fig. 4h). Therefore, while the DBD-C domain of Rfa1 promotes Dna2 recruitment, it is dispensable for nuclease stimulation. Under limiting concentrations, recruitment of Dna2 will likely be prerequisite for Dna2 to degrade DNA, however our experiments suggest that it may not be sufficient. The functions of RPA to promote Dna2 nuclease and recruit Dna2 are thus at least in part genetically separable.

In DNA end resection, RPA promotes the degradation of 5′-terminated ssDNA, while it inhibits 3′-ssDNA degradation, enforcing thus the correct polarity of resection. This specificity is likely dependent on the directional orientation of RPA when bound to ssDNA[13,39], and the strong physical interaction between Dna2 and Rfa1-F at the N-terminus of Rfa1[33]. When degrading the 5′-terminated strand, Rfa1-F would be the first RPA domain that Dna2 encounters on ssDNA, while the Rfa2-D would be the first encountered domain from the other side when degrading the 3′-terminated strand. We observed that RPA-SSL was indistinguishable from wild type RPA in its function to protect 3′-terminated ssDNA from DNA degradation by Dna2. In contrast, the low affinity ssDNA binders RPA-K494E and RPA-CCAA were slightly impaired in 3′-ssDNA protection (Fig. 4k and Supplementary Fig. 4i). The function of RPA to protect 3′-DNA from degradation therefore appears be a direct consequence of DNA binding of RPA and its polarity. When encountering RPA from the 3′ DNA side, a lack of specific physical and functional interactions makes RPA a block for Dna2 degradation, but only when bound tightly to ssDNA.

We conclude that the Rfa1 domains that mediate Dna2 recruitment do not match those required to promote the Dna2 nuclease, and the stimulation of the Dna2 nuclease is thus not a simple consequence of Dna2 recruitment to substrate DNA. We identify mutations within a helix of the N-terminal Rfa1 domain, leading to the RPA-SSL mutant, which is proficient to facilitate Dna2 recruitment and the protection of the 3′-terminated ssDNA, but it is strongly impaired in the stimulation of the Dna2 nuclease activity acting on 5′-terminated DNA.

**RPA-I14S and RPA-SSL mutants promote the Dna2-E675A helicase**. The Dna2 motor may not function as a helicase to unwind dsDNA in vivo, but rather as a ssDNA translocase. The motor facilitates movement of Dna2 along ssDNA, accelerating its degradation by the Dna2 nuclease domain[19,30,32]. We used the nuclease-deficient Dna2-E675A mutant as a tool to study the effect of the RPA variants on the Dna2 motor activity. We first employed oligonucleotide-based DNA, with 45-nt-long ssDNA arms and 48 base pairs of dsDNA. We observed that the RPA-K45E, RPA-I14S and RPA-SSL mutants were indistinguishable from wild type RPA to promote the helicase of Dna2-E675A, while the ssDNA-binding mutants RPA-K494E and RPA-CCAA were moderately impaired (Fig. 5a, b). Next, when monitoring the unwinding of 2.2-kbp-long dsDNA, RPA-SSL still promoted Dna2-E675A helicase as wild type RPA, while the RPA-K494E and RPA-CCAA did not support unwinding at all (Fig. 5c).

To define DNA unwinding by Dna2-E675A more quantitatively, we used single-molecule magnetic tweezers experiments[40]. To this point, 6.1-kbp-long DNA molecules were attached on one side to a surface of a fluidic cell, and on the other side, to a magnetic bead. An internal 5′-tailed ssDNA of 40 nt in length was left free to allow for RPA binding and the loading of Dna2-E675A (Fig. 5d). The DNA was first incubated with RPA to allow binding to the 5′-overhang. Then, the reaction was initiated by adding Dna2-E675A. The DNA unwinding was quantified from the apparent DNA length measured in this setup due to different specific extensions of double- and single-stranded DNA (Fig. 5d). Despite RPA being pre-bound to the 5′-overhang, no DNA unwinding by Dna2-E675A was detected without RPA in solution (Supplementary Fig. 5a). Upon supplementing free RPA into the solution, in addition to RPA pre-bound to the overhang, we observed long-range DNA unwinding at a mean rate of $44 \pm 3$ bp/s and a mean processivity $2.3 \pm 0.3$ kbp (Fig. 5e–g), similar to previous measurements[29]. Importantly, the RPA-I14S mutant supported DNA unwinding comparably as wild type RPA (Fig. 5h–j), in agreement with our bulk unwinding assays. We conclude that the RPA-I14S and RPA-SSL mutants are not affected in their function to promote the motor of Dna2.

**The DBD-A and DBD-B domains specifically promote the Dna2 helicase**. RPA promotes DNA unwinding by Dna2-E675A in a specific manner, beyond its capacity to recruit Dna2 to DNA (Figs. 1e, f and 2f). To define which regions of RPA are required to stimulate this unwinding activity, we tested the RPA fragments and subunits in unwinding assays. While heterotrimeric wild type RPA promoted the unwinding of 2.2-kbp-long dsDNA, the large subunit Rfa1 was incapable to do so (Fig. 5k and Supplementary Fig. 5b). This stands in contrast with the nuclease assays, where Rfa1 could replace the RPA heterotrimer. Unexpectedly, we found that truncations of the Rfa1 subunit, resulting in Rfa1-FAB, Rfa1-AB and Rfa1-ABC subunit combinations could readily promote DNA unwinding, in contrast to full-length Rfa1 (FABC) (Fig. 5k). Therefore, the minimal RPA region required to promote the Dna2-E675A helicase corresponds to the central Rfa1-AB domains, which typically do not associate with other proteins. Our results also indicated that the N-terminal Rfa1-F and C-terminal Rfa1-C can be inhibitory for DNA unwinding by Dna2-E675A, although we cannot exclude that some of these effects may be caused by instability of the Rfa1 fragments. We conclude that the Rfa1 domains required for long-range DNA unwinding are different from those required for DNA recruitment and the stimulation of the Dna2 nuclease.

We noted that the full-length Rfa1 subunit could still facilitate apparent DNA unwinding of short oligonucleotide-based DNA (Fig. 5l and Supplementary Fig. 5c), albeit less than heterotrimeric RPA, in accord with the conclusion that Rfa1 is insufficient to stimulate the Dna2 helicase function, unlike the nuclease function. We reasoned factors other than DNA unwinding per se likely affect the product formation, such as DNA reannealing or Dna2 recruitment. To define the function of RPA in promoting DNA unwinding more accurately, we turned again to the single-molecule experiments. In agreement with the bulk assays with long DNA, we failed to observe DNA unwinding by Dna2-E675A in the presence of Rfa1 (Fig. 5m, n and Supplementary Fig. 5b). In contrast, we observed DNA unwinding with Rfa1-FAB and Rfa1-ABC, highlighting once more the function of Rfa1-AB domains of Rfa1 to promote DNA unwinding by Dna2-E675A. The processivity and rates of DNA unwinding in conjunction with the tested RPA/Rfa1 variants were very similar to wild type, with one exception (Fig. 5m, n and Supplementary Fig. 5d–i). Unexpectedly, the rates of DNA unwinding by Dna2-E675A in the presence of Rfa1-FAB were even higher than with heterotrimeric RPA, suggesting again that the Rfa1-C subunit may be inhibitory for DNA unwinding.

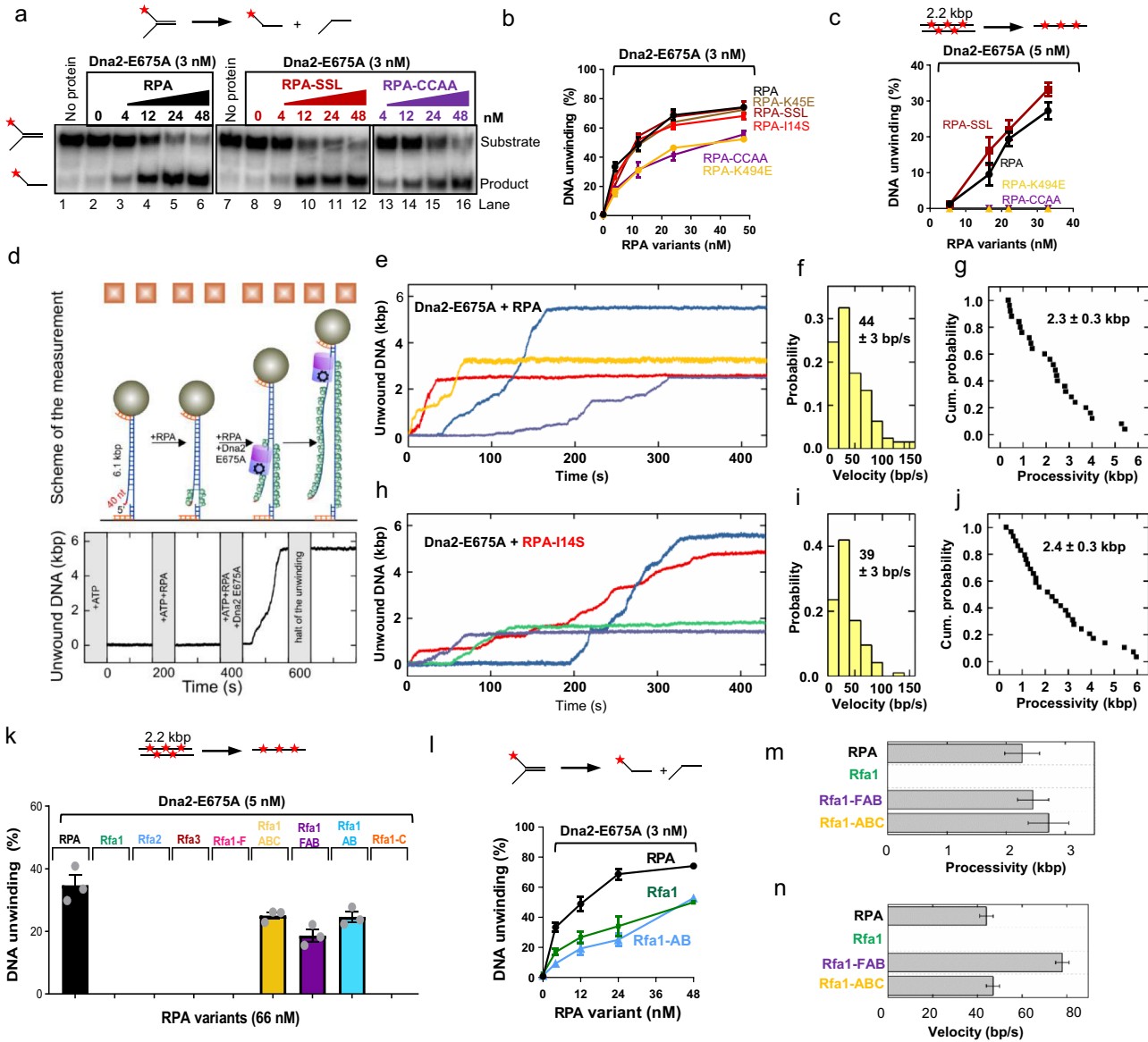

**Fig. 5 RPA-I14S and RPA-SSL mutants promote the Dna2-E675A helicase. a** Unwinding of a Y-shaped DNA substrate (45 nt ssDNA, 48 bp dsDNA 0.1 nM, in molecules) by Dna2-E675A in the presence of RPA variants and 50 mM KCl. The red asterisk indicates the position of the radioactive label. **b** Quantification of assays such as shown in panel **a**. Error bars, SEM; $n = 3$. **c** Quantification of 2.2-kbp-long dsDNA substrate (0.1 nM, in molecules) unwinding by Dna2-E675A in the presence of RPA variants. Error bars, SEM; $n = 3$. **d** A sketch of the employed magnetic tweezers assay including the DNA construct carrying a 40-nt-long 5′-ssDNA flap to allow loading of Dna2-E675A (top). DNA unwinding trajectory with sequential addition of reagents revealing that DNA unwinding was only observed after the addition of Dna2-E675A. **e** Representative trajectories of DNA unwinding events by Dna2-E675A (5 nM). The reaction was supplemented with RPA (20 nM). **f**, **g** Velocity histogram and cumulative probability distribution (shown as survival probability) of the processivity obtained from DNA unwinding trajectories of Dna2-E675A in presence of RPA. Error, SEM; $n = 25$. **h** Representative trajectories of DNA unwinding by Dna2-E675A (5 nM). The reaction was supplemented with RPA-I14S (20 nM). **i**, **j** Velocity histogram and cumulative probability distribution (shown as survival probability) of the processivity obtained from DNA unwinding trajectories of Dna2-E675A in presence of RPA-I14S. Error, SEM; $n = 29$. **k** Quantification of 2.2-kbp-long dsDNA substrate (0.1 nM, in molecules) unwinding by Dna2-E675A in the presence of RPA, subunits of RPA and fragments of Rfa1, as indicated. Error bars, SEM; $n = 3$. **l** Quantification of Y-shaped DNA substrate (45 nt ssDNA, 48 bp dsDNA 0.1 nM, in molecules) unwinding such as shown in Supplementary Fig. 5c. RPA is replotted as in panel **b** for reference. Error bars, SEM; $n = 3$. **m** Bar graph of processivities of DNA unwinding events by Dna2-E675A (5 nM) in the presence of RPA variants (20 nM). The distribution of the data is shown in Fig. 5g, j and Supplementary Fig. 5f, i. Error, SEM; $n = 25, 25, 30$ and $22$ for RPA, Rfa1, Rfa1-FAB and Rfa1-ABC. Representative traces are shown in Supplementary Fig. 5d, g. **n** Bar graph of velocities of DNA unwinding by Dna2-E675A (5 nM) in the presence of RPA variants (20 nM). The distribution of the data is shown in Fig. 5f, i and Supplementary Fig. 5e, h. Error, SEM; $n = 25, 25, 30$ and $22$ for RPA, Rfa1, Rfa1-FAB and Rfa1-ABC. Representative traces are shown in Supplementary Fig. 5d, g.

To understand the role of Rfa1-AB in promoting DNA unwinding by Dna2-E675A, we applied structure modeling. Based on sequence analysis and published mammalian DNA2 and RPA structures[4,22], we performed 3D docking to model the potential Dna2 and Rfa1-AB interaction (Supplementary Fig. 6a). In the obtained model, a conserved and exposed tyrosine residue (Y193) of Rfa1-A could act as an anchor to interact with an apolar site on Dna2, near the region where the 3′-end of the threaded DNA lies in DNA2 (Fig. 6a and Supplementary Fig. 6a). To block this potential interaction, we designed Rfa1-AB-Y193A and Rfa1-AB-Y193D mutants to disrupt the tyrosine, and Rfa1-AB-S191K to create a steric hindrance at the interaction site. We then expressed and purified the Rfa1-AB variants (Fig. 6b and Supplementary Fig. 6b). As indicated in the structure model, the mutations are located on the surface of Rfa1-AB on the opposite side of the ssDNA binding site. The mutations consequently did not notably affect the binding of Rfa1-AB to ssDNA (Fig. 6b, c, and Supplementary Fig. 6c).

We next analyzed the Rfa1-AB mutants for their capacity to promote long-range DNA unwinding by Dna2-E675A. We observed that all mutants exhibited reduced DNA unwinding in conjunction with Dna2-E675A compared to wild type Rfa1-AB. In particular, the Rfa1-AB-Y193D mutant was ~2.5-fold less efficient in promoting Dna2-E675A unwinding activity in assays with various protein concentrations, as well as in a kinetic experiment (Fig. 6d and Supplementary Fig. 6d), validating the docking model. Next, we combined the S191K and Y193D mutations in the heterotrimeric RPA, creating the SKYD mutant (Fig. 6e and Supplementary Fig. 6e). Although RPA-SKYD bound ssDNA similarly as wild type (Fig. 6f and Supplementary Fig. 6f), it notably reduced DNA unwinding by Dna2-E675A (Fig. 6g, h). Following the defects of RPA-SKYD in promoting the helicase activity of Dna2-E675A, we wanted to characterize if it has similar effects on the capacity of Dna2-E675A to hydrolyze ATP. To this point, we used a spectrophotometric assay, which measures ATPase activity based on a reaction coupled to the oxidation of NADH. We compared the ATPase activity of Dna2-E675A in the presence of the various RPA point mutants[29]. We observed that the apparent $k_{cat}$, which is the apparent ATP turnover number, for Dna2-E675A was stimulated approximately 2-fold in the presence of RPA wild type and RPA-SSL. However, the RPA-SKYD mutant did not stimulate the ATPase of Dna2-E675A (Fig. 6i). Therefore, residues located on the outer surface of the major DNA-binding domain A of RPA, around Y193, are not involved in ssDNA binding, but are likely required to transiently interact with Dna2 to promote its motor function[22]. The RPA-SKYD mutant only weakly reduced nucleolytic degradation of 2.2-knt-long ssDNA (Fig. 6j and Supplementary Fig. 6g), in contrast to RPA-SSL, which brought about a dramatic DNA degradation defect, in accord with Fig. 4f, g that utilized oligonucleotide-based DNA. These experiments confirm that Rfa1 residues within DBD-F primarily promote the Dna2 nuclease, while Rfa1 residues within DBD-A promote the motor activity of Dna2.

**RPA-SSL and SKYD mutants are impaired in stimulating DNA end resection**. It has been established that Dna2, Sgs1 and RPA comprise a minimal protein complex capable of DNA end resection in vitro[26,27]. Both nuclease and motor activities of Dna2 are required for maximal DNA end resection capacity[30,32]. To verify the importance of the newly-identified RPA variants for a physiologically relevant process such as DNA end resection, we explored the capacity of the RPA variants to function with Dna2 and Sgs1 in reconstituted resection assays. We observed that RPA-SSL and RPA-SKYD mutants both showed notably reduced capacity to resect DNA, despite their proficient ssDNA binding affinity (Fig. 7a, b). Additionally, these mutants were comparable in terms of their capacity to stimulate the Sgs1 helicase, suggesting that the defects of these mutants in resection are majorly due to the impaired interplay with the Dna2 helicase-nuclease (Supplementary Fig. 7a, b). We note that we failed to construct the RPA-SSL and RPA-SKYD mutants in yeast cells, suggesting a likely lethal phenotype, in agreement with already notably reduced growth rate of RPA-I14S[36]. Our results together confirm that RPA, beyond its ssDNA binding functions, has specific roles to recruit and activate the Dna2 helicase-nuclease activities to promote DNA end resection in DSB repair.

## Discussion

Here we define the functional interactions between the eukaryotic ssDNA binding protein RPA and the *S. cerevisiae* Dna2 nuclease-helicase, which participates in diverse DNA metabolic processes including replication and recombination. RPA is known to specifically promote Dna2 recruitment to overhanged or flapped DNA, and promote both nuclease and helicase activities of Dna2[16,21,22,26,27,29]. Human mitochondrial SSB, which resembles prokaryotic ssDNA binders, instead inhibits all Dna2 activities, which underlines the specific nature of the interaction between cognate yeast RPA and yeast Dna2. Dna2 was shown to bind the large Rfa1 subunit of RPA[33], and the stimulatory activity of RPA on the enzymatic activities of Dna2 was thought to be a consequence of enhanced recruitment. Our data show that the stimulation of the Dna2 nuclease and helicase activities is more direct than anticipated. We demonstrate that RPA has the capacity to specifically promote both enzymatic activities of Dna2 in addition to its recruitment function.

Previously, Dna2 was shown to physically interact with the N-terminal (Rfa1-F) and less efficiently with the additional Rfa1-A, -B and -C subunits[22,33]. We show that both Rfa1-F and Rfa1-C domains are required for efficient recruitment of Dna2 to overhanged DNA when coupled with the central Rfa1-AB domains (Fig. 7c). In contrast, the Rfa1-AB construct alone, which binds DNA with a high affinity, fails to recruit Dna2. The contribution of Rfa1-F and Rfa1-C domains towards recruitment of Dna2 is likely dependent on the described physical interactions[22,33]. Whether RPA remains a part of the nucleoprotein complex upon Dna2 recruitment has been controversial, likely due to difficulties interpreting the mobilities of protein-bound complexes in electrophoretic shift assays[16,21,22]. Using mass photometry, we observed that upon the recruitment of Dna2, RPA remains a part of the nucleoprotein complex as an intermediate. RPA is thus positioned to be capable to stimulate also the enzymatic activities of Dna2 that occur downstream of recruitment (Fig. 7c).

We show that Rfa1 alone can promote the Dna2 nuclease equally well as heterotrimeric RPA. The Rfa1-C domain appears inhibitory in this regard. On the other hand, the N-terminal Rfa1-F domain, coupled with Rfa1-AB, has a critical function to promote cleavage of 5′-terminated ssDNA, although it does not contribute to ssDNA binding (Fig. 7d and Supplementary Fig. 7c). We identify the RPA-SSL variant with mutations in a conserved helix embedded in the N-terminal Rfa1-F domain. The mutations likely affect the local structure that extends to residues exposed to the surface. The resulting mutant is proficient in DNA binding and Dna2 recruitment, but strongly impaired in the stimulation of DNA degradation by Dna2, as observed with both oligonucleotide-based and plasmid-length DNA substrates. Therefore, recruitment of Dna2 and stimulation of its nuclease activity are genetically separable.

As with the Dna2 nuclease, RPA promotes the Dna2 helicase function beyond recruitment. The stimulation of DNA

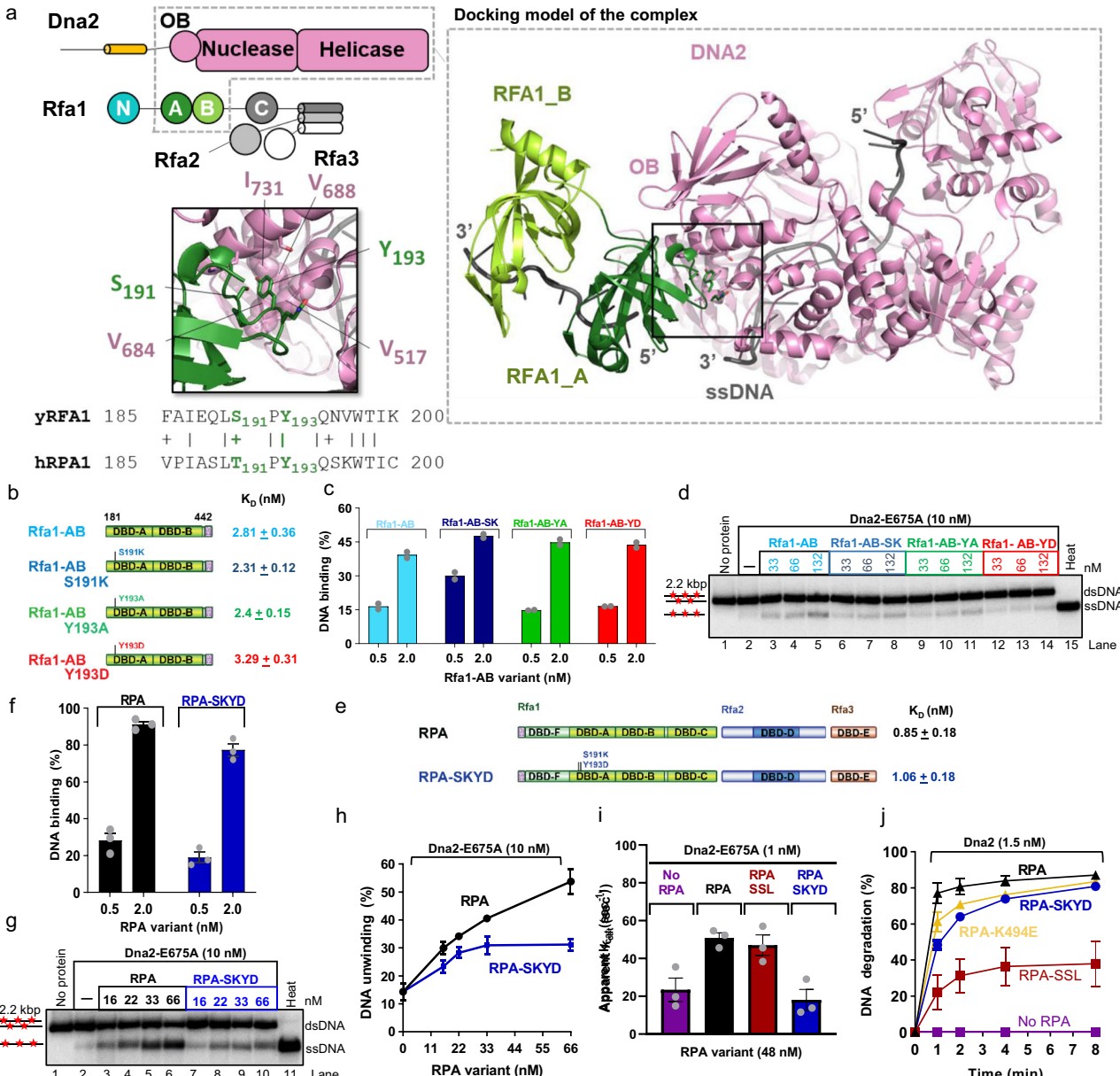

**Fig. 6 The DBD-A and DBD-B domains of Rfa1 specifically promote the Dna2 helicase. a** A schematic representation of the domain organization in Dna2 and RPA, highlighting with a dashed gray box the domains used as inputs of the free docking simulation. Right, most likely structural model obtained after free docking with the InterEvDock2 server. Dna2 nuclease-helicase domain and the AB module of Rfa1 are represented in pink and green, respectively, ssDNA is in dark gray. Lower left panel focuses on the interaction region of Rfa1 (in green) with Dna2 (in pink), highlighting the conserved residues S191 and Y193 of Rfa1. Y193 is predicted to anchor in an apolar pocket exposed at the surface of Dna2 formed by V517, V684, V688 and I731. Bottom, pairwise sequence alignment between *S. cerevisiae* Rfa1 and human RPA1 sequences in the region 185–200 highlights the conservation of the Y193 residue. **b** A schematic representation of the primary structure of wild type Rfa1-AB and point mutants. Rfa1-AB is again shown as reference. $K_D$, concentration of the respective Rfa1-AB variant resulting in 50% binding to ssDNA (93 nt, 0.1 nM, in molecules) such as shown in Supplementary Fig. 6c. Error, range; $n = 2$. **c** Quantitation of Rfa1-AB variants binding to ssDNA (93 nt, 0.1 nM, in molecules) as shown in Supplementary Fig. 6c. Rfa1-AB is replotted as in Fig. 3e for reference. Bars show range; $n = 2$. **d** Representative experiments showing unwinding of 2.2-kbp-long dsDNA (0.1 nM, in molecules) by Dna2-E675A in the presence of Rfa1-AB variants. Red asterisks indicate random radioactive labels on the DNA. The experiment was performed three times with similar results. **e** Primary structure of the RPA-SKYD mutant. Wild type RPA is again shown as a reference. $K_D$, concentration of the respective Rfa1-AB variant resulting in 50% binding to ssDNA (0.1 nM, in molecules, 93-nt-long) such as shown in Supplementary Fig. 6f. Error, SEM; $n = 3$. **f** DNA binding by RPA-SKYD to ssDNA (93 nt, 0.1 nM, in molecules) as shown in Supplementary Figs. 3c and 6f. RPA is replotted as in Fig. 3b for reference. Error bars, SEM; $n = 3$. **g** Representative experiments showing unwinding of 2.2-kbp-long dsDNA (0.1 nM, in molecules) by Dna2-E675A in the presence of wild type RPA or RPA-SKYD. **h** Quantification of helicase assays such as shown in panel **g**. Error bars, SEM; $n = 3$. **i** Apparent ATP turnover number and its dependence on various RPA variants (48 nM) in the presence of single-stranded DNA (50 nt, 1 μM, in nucleotides). The reactions contained Dna2 E675A (1 nM) and ATP (1 mM). Error bars, SEM, $n = 3$. **j** Quantification of nuclease assays such as shown in Supplementary Fig. 6g showing kinetics of degradation of 2.2-knt-long ssDNA (0.3 nM) by Dna2 in the presence of RPA variants (50 nM) and with 100 mM KCl. Error bars, SEM; $n = 3$.

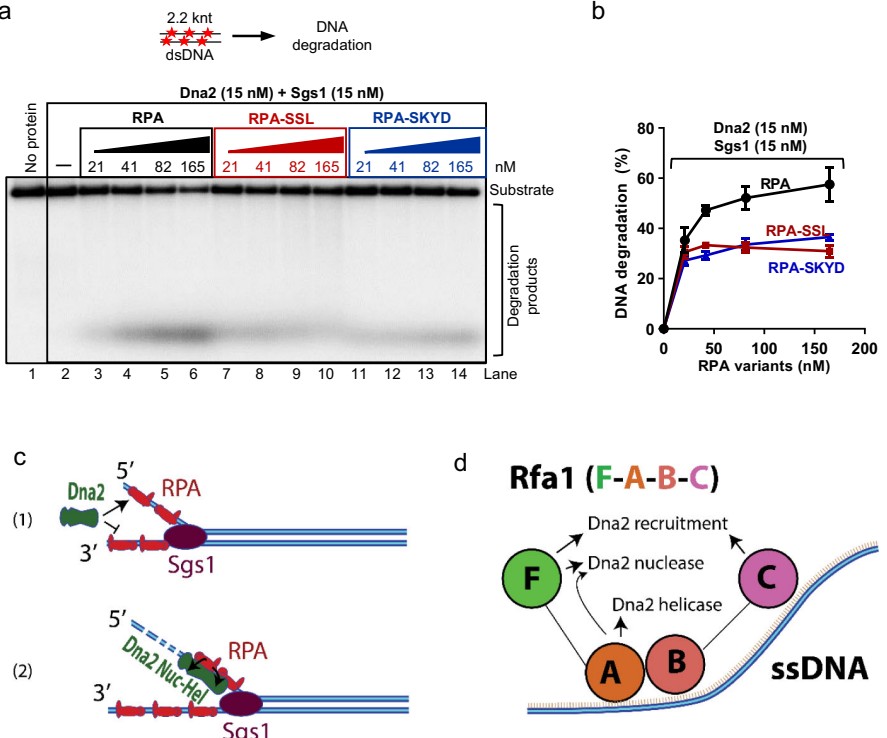

**Fig. 7 RPA-SSL and SKYD mutants are impaired in DNA end resection. a** A representative experiment to monitor DNA end resection by Sgs1, Dna2, and increasing concentrations of yeast RPA variants using 2.2-knt-long dsDNA (0.5 nM, in molecules). The reaction buffer contained 100 mM NaCl. Red asterisks indicate random radioactive labels on the DNA. **b** Quantification of overall substrate utilization from experiments such as shown in panel **a**. Error bars, SEM; $n = 3$. **c** A model showing that RPA first recruits Dna2 to 5′-overhanged DNA (1). Subsequently, RPA remains a component of the nucleoprotein complex and stimulates both nuclease (Nuc) and ATPase-driven translocase (Hel) activities of Dna2 (2). **d** A model depicting domains of the RPA large subunit, Rfa1, and their involvement in ssDNA binding, recruitment of Dna2 and stimulation of nuclease and helicase activities of Dna2, respectively.

unwinding by RPA also cannot be explained by non-specific prevention of DNA reannealing. When nuclease-deficient Dna2-E675A was prebound to DNA, making recruitment irrelevant, RPA promoted DNA unwinding much more than mtSSB. Under these conditions, mtSSB was expected to prevent reannealing equally well as RPA. Likewise, in single-molecule assays, RPA prebound to the ssDNA overhang did not support long-range DNA unwinding by Dna2-E675A in absence of additional free RPA. These experiments collectively suggest that RPA is required for the stimulation of the motor activity of Dna2, and becomes essential for the unwinding of long DNA substrates. Surprisingly, the RPA-SSL mutant that largely failed to promote the Dna2 nuclease activity stimulated DNA unwinding by Dna2-E675A equally well as wild type RPA, showing that that the stimulation of the nuclease and helicase activities involves different regions of RPA. In contrast to recruitment and nuclease stimulation, Rfa1 alone could not promote long-range DNA unwinding by Dna2. Unexpectedly, the Rfa1-AB, as well as the Rfa1-ABC and Rfa1-FAB fragments activated the helicase activity of Dna2-E675A almost as well as the heterotrimeric RPA. Therefore, when using only Rfa1, the N-terminal Rfa1-F and C-terminal Rfa1-C domains were surprisingly inhibitory for long-range DNA unwinding by Dna2-E675A. Structure modeling and three-dimensional molecular docking identified a potential interaction site between the surface of Rfa1-A and Dna2. We could validate this model by mutating a conserved tyrosine residue in Rfa1-AB (Y193), which was important for the ability of the Rfa1 fragment to promote DNA unwinding. Consequently RPA-SKYD, which carried the S191K mutation in addition to Y193D within the RPA heterotrimer, was proficient to bind ssDNA, but impaired to stimulate DNA unwinding and as well as Dna2-E675A ATPase

activity. These experiments showed that Dna2 also interacts with the Rfa1-A domain, which is quite unusual for RPA-interacting proteins.

Together, our data show that a certain level of ssDNA binding activity by RPA is prerequisite for Dna2 stimulation. However, the full complement of ssDNA binding capacity of RPA may not be necessary, as mutants with reduced ssDNA binding retained Dna2 stimulation. Rather, the stimulatory activity of RPA depends additionally on specific interactions between distinct RPA domains and Dna2. Most surprisingly, we found that different RPA domains are responsible for stimulating the diverse Dna2 functions including DNA binding, DNA nuclease and DNA helicase activities, showing that the functions of Dna2 are intertwined with RPA much more than previously assumed (Fig. 7d and Supplementary Fig. 7). We show that both RPA-SSL (strongly impaired to promote the Dna2 nuclease, but not the helicase activity) and RPA-SKYD (strongly impaired to promote the Dna2 helicase, but with only a weak effect on the nuclease activity) showed reduced capacity to function in DNA end resection together with Dna2 and Sgs1, underpinning the importance of both enzymatic activities of Dna2 in DSB repair.

## Methods

**Cloning, expression, and purification of recombinant proteins**. To prepare the expression construct for his-tagged full-length trimeric yeast *S. cerevisiae* RPA, a sequence coding for 6x-his tag was introduced to the 5′-end of *RFA1* in p11d-sctRPA vector (a kind gift from M. Wold, University of Iowa)[41]. To this point, primers AA07 and AA08 were annealed and cloned into NcoI site in p11d-sctRPA, creating p11d-his-sctRPA. See Supplementary Tables S1 and S2 for the list and sequences of all primers used in this study. All point mutants of full-length trimeric yeast RPA were introduced into this construct using mutagenic primers and QuikChange II-XL site-directed mutagenesis kit (Agilent Technologies) according

to manufacturer's recommendations. To prepare Rfa1, Rfa2, Rfa3, Rfa1-F, Rfa1-FAB, Rfa1-ABC and Rfa1-AB expression constructs, the respective DNA sequences were amplified by PCR using the corresponding primers (Supplementary Tables S1 and S2) and ligated into pET21 vector. These constructs contained a T7-tag at the N- and 6x-his tag at the C-terminus. Point mutants of Rfa1 and Rfa1-AB were also introduced into pET21-T7-Rfa1-his and pET21-T7-Rfa1-AB-his, respectively, using mutagenic primers as described above for the trimeric mutants.

All yeast recombinant trimeric full-length his-tagged wild type RPA and mutants (I14S, K45E, K494E, CCAA, SSL and SKYD) were expressed in BL21-DE3 pLysS E. coli cells. Transformed cells were grown as a pre-culture at 30 °C for 16 h without shaking. This pre-culture was inoculated into 1 l of LB medium and the cells were grown at 37 °C (200 rpm) until OD$_{600}$ reached ~0.6, and protein expression was induced with 0.4 mM isopropyl b-D-1-thiogalactopyranoside (IPTG) for 16 h at 18 °C. Bacterial pellet was obtained by centrifugation at 2500 × g for 15 min at 4 °C, the cells were washed once with SD Buffer (10 mM Tris-HCl pH 8.0, 150 mM NaCl, 1 mM ethylenediaminetetraacetic acid [EDTA]), snap-frozen and stored at −80 °C. Rfa1, Rfa2, Rfa3, Rfa1-ABC, Rfa1-C and point mutants of Rfa1 (I14S and K494E) were expressed as described for the full-length his-tagged RPA. Rfa1-F, Rfa1-FAB, Rfa1-AB, and all point mutants of Rfa1-AB (S191K, Y193A and Y193D) were prepared similarly, except that the cells were grown at 37 °C for 3 h after IPTG induction.

Untagged yeast RPA was expressed from p11d-sctRPA in BL21-DE3 pLysS Escherichia coli and purified using ÄKTA pure (GE Healthcare) with HiTrap Blue HP, HiTrap desalting and HiTrap Q HP chromatography columns (all GE Healthcare)[42].

Trimeric his-tagged wild type RPA and point mutants (I14S, K45E, K494E, CCAA, SSL and SKYD) were purified using nickel-nitrilotriacetic acid (Ni-NTA agarose [Qiagen]) and HiTrap Q HP column chromatography. Frozen pellets were resuspended in lysis buffer (50 mM Tris-HCl pH 7.5, 1 mM phenylmethylsulfonyl fluoride [PMSF], 2 mM β-mercaptoethanol [β-ME], 10% glycerol, 150 mM KCl, 20 mM imidazole, 0.1% NP40, 1:300 protease inhibitor cocktail [Sigma, P8340]) and sonicated. The lysate was clarified by centrifugation at 51,000 × g for 0.5 h and incubated with Ni-NTA resin for 1 h batchwise at 4 °C. The resin was washed 2 times batchwise with 40 ml wash buffer I (lysis buffer without protease inhibitor cocktail) and then 6 times the column volume with the same buffer on a disposable column (10 ml, Thermo Fisher) by gravity flow. The resin was further washed with 9 column volumes of wash buffer II (wash buffer I supplemented with 50 mM imidazole). The resin was then washed with 0.5 column volume of wash buffer II supplemented with 5 mM MgCl$_2$ and 1 mM ATP. This was followed by elution with elution buffer (wash buffer II containing 300 mM imidazole). The eluate was then diluted 8-fold with dilution buffer (50 mM HEPES-KOH pH 7.5, 10% glycerol, 2 mM β-ME, 94 mM KCl) and loaded on a pre-equilibrated HiTrap Q column. The loaded column was washed first with dilution buffer containing 100 mM KCl, and then 170 mM KCl, before finally eluting the protein in a salt gradient from 170 mM to 800 mM KCl under identical buffer conditions. The eluate fractions were analyzed on a 12% polyacrylamide gel to identify the best ones, which were pooled, aliquoted, snap frozen in liquid nitrogen and stored at −80 °C. Recombinant Rfa2 and Rfa3 were also purified as above, except that the wash buffer I contained 1 M KCl. Recombinant Rfa1, Rfa1-ABC and Rfa1-C were purified using the same procedure as Rfa2 and Rfa3, but diluted after the first Ni-NTA step and loaded onto the HiTrap Q column at pH 8.5. These fragments were further washed with wash buffers with pH 8.5 and eluted as described above in buffers with pH 7.5. Recombinant Rfa1-F, Rfa1-AB and Rfa1-FAB constructs were purified using Ni-NTA and HiTrap-Heparin column, since they did not bind to the HiTrap Q column. The Ni-NTA purification step was done as described for the fragments (Rfa1, Rfa1-ABC and Rfa1-C) above, but washed with wash buffer II containing 40 mM imidazole. The heparin step was performed as described for HiTrap Q purification of Rfa1, Rfa1-ABC and Rfa1-C. Point mutants of Rfa1 (I14S and K494E) and Rfa1-AB (S191K, Y193A and Y193D) were purified using Ni-NTA only following the above described procedures for their corresponding wild type proteins. These final protein samples were dialyzed into storage buffer (50 mM Tris-HCl pH 7.5, 2 mM β-ME, 10% glycerol, 1 mM PMSF and 100 mM NaCl).

Recombinant S. cerevisiae wild type and nuclease-dead Dna2 variants were expressed from a modified pGAL:DNA2 vector that contained N-terminal FLAG and HA-tags and a C-terminal 6x-his tag, in the S. cerevisiae strain WDH668[43,44]. Yeast cells (4 l) were grown to approximately OD$_{600}$ ~0.6 in a standard synthetic medium lacking uracil, and supplemented with glycerol (3%, vol/vol) and lactic acid (2%, vol/vol) as carbon sources. Expression of Dna2 was induced with galactose (2%) for 6 h and the cells were lysed in a freezer mill (Spex Sample Prep). All Dna2 variants were purified by Ni-NTA and FLAG (Sigma, A2220) affinity chromatography[29]. Human mitochondrial SSB was expressed from pMALT-P-mtSSB in 1 l of BL21-DE3 pLysS E. coli cells upon induction with 0.4 mM IPTG for 3 h at 37 °C, and purified using amylose affinity resin (New England Biolabs) and HiTrap Blue HP[28]. Recombinant S. cerevisiae Sgs1 protein was expressed from the pFB-MBP-Sgs1-his vector in Spodoptera frugiperda 9 (Sf9) insect cells by using Bac-to-Bac baculovirus expression system (Invitrogen) and purified using Amylose and Ni-NTA affinity chromatography[45].

**DNA substrate preparation**. For DNA-binding experiments, ssDNA oligonucleotide (93-nt-long, X12-3HJ3)[28] was labeled at the 3′ terminus with [α-$^{32}$P] dCTP

(Perkin Elmer) and terminal transferase (New England Biolabs), according to standard protocols. Unincorporated nucleotides were removed using Micro Bio-Spin P-30 Gel Columns (Bio-Rad). For oligonucleotide-based nuclease assays and DNA recruitment assays, either of two 5′-overhanged DNA substrates (45 nt ssDNA and 48 bp dsDNA, oligonucleotides X12-3 HJ1S and X12-3 TOPL; or 30 nt ssDNA and 31 bp dsDNA, oligonucleotides PC92 and X12-4SC, as indicated) were used[28,29]. Recruitment measurements using mass photometry used a DNA substrate with a 25 nt 5′-overhang and a 48 bp dsDNA region (assembled from oligonucleotides X12-3 HJ1S[28] and truncated X12-3 TOPL [TGCTAGGACATGCTGTCTAGAGACTATCGCGACTTACGTTCC ATCGCTAGG TTATTTTTTTTTTTTTTTTTTTTT]). Protection assays with 3′-over-hanged DNA were performed using 45 nt ssDNA, 48 bp dsDNA substrate prepared by annealing X12-3 HJ2Sb and X12-3HJ3[28], with the label on the 3′ terminus. DNA end resection assays were performed using 2.2-kbp-long PCR-based dsDNA substrate[31], which was randomly labeled by incorporating [α-$^{32}$P] dCTP during PCR. This substrate was heat-denatured to perform the kinetic experiments to study the ssDNA nuclease function of Dna2. For helicase assays, Y-structured (oligonucleotides X12-3HJ3 and X12-3TOPL), 5′-overhanged (oligonucleotides PC92 and X12-4SC) and double-stranded (oligonucleotides X12-3 and X12-4C) DNA substrates were used[28,29]. For ATPase assays, single-stranded X12-3 oligonucleotide was used. The dsDNA construct for the magnetic tweezers assays was prepared as described[28,34]. A 6.6 kbp dsDNA fragment was excised from pNLrep plasmid[46], using the restriction enzymes BamHI and BsrGI. Simultaneously, a 63 nt gap was formed by utilizing the Nt.BbvCI nicking enzyme. The gap enabled annealing of a 65-nt-long oligonucleotide with a 25 nt complementary sequence and a 40 nt polythymidine 5′ tail. Subsequently, 0.6-kbp-long dsDNA handles containing either multiple biotin or digoxigenin modifications were ligated to each end of the 6.6 kbp fragment to support specific anchoring to the streptavidin-coated magnetic bead and the antidigoxigenin-coated glass coverslip.

**Electrophoretic mobility shift assays**. DNA-binding reactions (15 μl volume) were carried out in 25 mM Tris-acetate pH 7.5, 5 mM magnesium acetate, 200 mM KCl, 1 mM dithiothreitol (DTT), 100 μg/ml bovine serum albumin (BSA, New England Biolabs), ssDNA substrate (oligonucleotide X12-3HJ3[28], as indicated, 0.1 nM, in molecules). Proteins were added and incubated at 30 °C for 30 min. Loading dye (5 μl; 50% glycerol, bromophenol blue) was added to the reactions and the products were separated on 6% polyacrylamide gels (ratio acrylamide:bisacrylamide 19:1, Bio-Rad) in TAE buffer (40 mM Tris, 20 mM acetic acid and 1 mM EDTA) at 4 °C. The gels were dried on 17 CHR (Whatman), exposed to a storage phosphor screen (GE Healthcare) and scanned by a Typhoon Phosphor Imager (FLA 9500, GE Healthcare). Binding of Dna2-E675A to the Y structure (oligonucleotides X12-3HJ3 and X12-3TOPL, 0.1 nM, in molecules) was performed under identical conditions with 2 mM magnesium acetate and 1 mM ATP but without added salt.

**Nuclease assays**. Unless indicated otherwise, all oligonucleotide-based nuclease assays (15 μl volume) were carried out in 25 mM Tris-acetate pH 7.5, 2 mM magnesium acetate, 1 mM ATP, 1 mM DTT, 0.1 mg/ml BSA, 0.08 units/μl pyruvate kinase, 1 mM phosphoenolpyruvate, using long 5′-overhanged DNA substrate (oligonucleotides X12-3 HJ1S and X12-3 TOPL or PC92 and X12-4SC, as indicated, 1 nM, in molecules). Proteins were added and incubated at 30 °C for 30 min. Reactions were terminated with stop buffer (0.5 μl 10% [w/vol] sodium dodecyl sulfate [SDS] and 0.5 μl 0.5 M EDTA) and 0.5 μl proteinase K (14–22 mg/ml, Roche), and incubated for 30 min at 50 °C. Terminated reaction mixtures were mixed with an equal volume of loading dye (95% formamide, 20 mM EDTA, 1 mg/ml bromophenol blue) and separated by electrophoresis in 15% denaturing polyacrylamide gels (acrylamide:bisacrylamide, 19:1, Bio-Rad) in TBE (89 mM Tris, 89 mM boric acid, 2 mM EDTA) buffer. The gels were fixed for 30 min at room temperature in 40% methanol, 10% acetic acid and 5% glycerol; dried on 3MM paper (Whatman) and processed as described above. 3′-overhanged DNA substrate protection assays were performed under identical conditions, except that the buffer was supplemented with 100 mM NaCl. Kinetic experiments to study the degradation of 2.2-knt-long ssDNA (0.3 nM, in molecules) by Dna2 were performed using boiled 2.2-kbp-long dsDNA substrate (0.15 nM, in molecules) under identical reaction buffer conditions, except being supplemented with 100 mM KCl and separated by electrophoresis in 15% denaturing polyacrylamide gels.

**Helicase assays**. Unless indicated otherwise, all oligonucleotide-based helicase assays (15 μl volume) with Dna2-E675A were performed in the reaction buffer used for the nuclease assays supplemented with 50 mM KCl, using Y-shaped DNA as substrate (oligonucleotides X12-3HJ3 and X12-3TOPL or PC92 and X12-4SC, as indicated, 0.1 nM, in molecules). Recombinant proteins were added as indicated. The reactions were incubated at 30 °C for 30 min and stopped using 5 μl 0.2% stop buffer containing 0.2% SDS, 150 mM EDTA and 30% glycerol, and 1 μl proteinase K (14–22 mg/ml, Roche), and incubated at 37 °C for 10 min. To avoid re-annealing of the substrate, the stop solution was supplemented with a 20-fold excess of the unlabeled oligonucleotide with the same sequence as the $^{32}$P-labeled one. The products were separated by 10% polyacrylamide gel electrophoresis in TBE buffer, dried on 17 CHR chromatography paper (Whatman) and analyzed as described above. Plasmid based helicase assays (15 μl volume) were performed in the same way, except 2.2-kbp-long DNA substrate (0.1 nM, in molecules) was used. The

products were separated by 1% agarose (Sigma, A9539) gel electrophoresis in TAE buffer, squeezed, dried on DE81 paper (Whatman) and analyzed.

**ATPase assays.** ATPase assays with Dna2-E675A were performed in 25 mM Tris-acetate pH 7.5, 1 mM magnesium acetate, 1 mM DTT, 0.1 mg/ml BSA, 1 mM ATP and 1 mM PEP, 0.025 units/μl pyruvate kinase, 0.025 units/μl L-lactic dehydrogenase (Sigma), with ssDNA as a co-factor (1 μM, in nucleotides, oligonucleotide X12-3). Recombinant proteins were added as indicated. The samples were mixed on ice and the absorbance data were collected as a function of time using a spectrophotometer (Cary 60 UV-Vis, Agilent Technologies) equipped with a temperature controller at 30 °C using the Cary WinUV Kinetics application[47].

**DNA recruitment assays.** The assays were performed as the DNA-binding assays with short or long 5′-overhanged DNA substrate (1 nM, in molecules), except that the reactions contained 150 mM NaCl and the DNA was always precoated with the respective RPA variant on ice for 5 min. The reactions performed with wild type Dna2 contained 3 mM EDTA instead of 5 mM magnesium acetate to inactivate the nuclease. The products were separated on 4% polyacrylamide gels and analyzed as described above.

**DNA end resection assays.** DNA end resection assays (15 μl volume) were performed in 25 mM Tris-acetate pH 7.5, 2 mM magnesium acetate, 1 mM ATP, 1 mM DTT, 0.1 mg/ml BSA, 80 U/ml pyruvate kinase, 1 mM phosphoenolpyruvate, 100 mM NaCl and randomly labeled 2.2-kbp-long double-stranded DNA substrate (0.5 nM, in molecules). The products were separated by 1% agarose gel electrophoresis in TAE buffer, dried on DE81 paper (Whatman) and analyzed.

**Docking simulations to model the interaction between RPA1_AB domains and DNA2.** To generate a model of the assembly between the N-terminal domain of Rfa1 and the small region of Dna2 upstream of the OB fold, a structure of the complex between the orthologous subunits in human (PDB:5EAY) was used as template to produce a homology model using rosettaCM standard protocol[48] (Fig. 4a). To model the interaction between the domains A and B of Rfa1 with Dna2, no structure of homologous complex was available. Consequently, a rigid-body free docking simulation was performed using the InterEvDock2 server[49], which takes into account the physicochemical nature of protein surfaces and co-evolutionary information. InterEvDock2 computes a consensus between 3 complementary scores, Frodock[50], SOAP-PP[51] and InterEvScore[52] to identify the most likely interfaces (http://bioserv.rpbs.univ-paris-diderot.fr/services/InterEvDock2/). The free docking strategy used human rather than yeast proteins because it would have required to use homology models as inputs of the free docking protocol likely decreasing docking performance. Docking simulations were performed using the X-ray structures of human RPA1 (PDB:1JMC) and DNA2 (PDB:5EAN). The input structures were the crystal structure of the tandem OB domains A and B of human RPA70 and the crystal structure of human DNA2 (Fig. 6a). The docking protocol was performed following the server standard protocol[53,54]. The co-alignments between both DNA2 and RPA1 partners automatically generated by the server contained 101 sequences belonging to species ranging from metazoans to fungi. From the results archive, the 50 best decoys of every 3 scores (Frodock, SOAP-PP, InterEvScore) used in the consensus selection of the docking models were considered. There were two major regions at the surface of DNA2 involved in the most likely docked models (Supplementary Fig. 6a). The first surface patch involved a surface close to the 3′ end of the ssDNA molecule co-crystallized with DNA2 and to the N-terminal OB-fold of DNA2. Since it had previously been shown that the OB domain in DNA2 contributes to the binding of RPA[22], a second docking simulation was run to further sample around this first surface patch. A second docking run was constrained so as to consider only models for which residues D111, D114 and F115 of Dna2 (in the OB domain) are at less than 15 Å from any atom of RPA1. The Top5 models generated in the second run were selected and relaxed using Rosetta[55] to remove steric clashes through using a standard relax protocol under native coordinate constraints and using the beta_nov15 scoring function. The model scored in rank 3 by InterEvScore was selected based on the calculated interface energy and visual inspection of the interface. This model was converted into a model of the complex between the S. cerevisiae Dna2 and Rfa1-AB partners using comparative modeling by RosettaCM. The resulting model was used to design a disruptive mutant on Y193 and S191 likely to prevent the Rfa1-AB module to interact with DNA2 while keeping its affinity for ssDNA.

**Mass photometry characterization of protein complexes.** Mass photometry measurements were performed on a OneMP mass photometer (Refeyn Ltd). To prepare the measurements, borsilicate microscope coverslips (No. 1.5 H thickness, 24 ×50 mm, VWR) were cleaned by sequential sonication in Milli-Q-water, iso-propanol and Milli-Q-water followed by drying under a stream of clean nitrogen. Subsequently, silicone gaskets (CultureWellTM Reusable Gasket, Grace Bio-Labs) were placed on the cleaned coverslips to create well-defined wells for sample delivery. Prior to each measurement, RPA, wild type Dna2 and/or the DNA substrate with a 25 nt-long 5′ ssDNA overhang (see DNA substrate preparation) were pre-mixed in the measurement buffer (25 mM Tris-acetate pH 7.5, 150 mM NaCl, 3 mM EDTA) at twice the final concentration and allowed to incubate for 15 min at room temperature. For mass measurements, gaskets were filled with 10 μl measurement buffer to allow focusing the microscope onto the coverslip surface. Subsequently, 10 μl of the pre-mixed protein complexes were added into the gasket providing final concentrations of each protein species of 20–80 nM (see Fig. 2d for exact concentrations). Sample binding to the coverslip surface was monitored for 120 sec using the software AcquireMP (Refeyn Ltd, Version 2.3.0). Data analysis was performed using DiscoverMP (Refeyn Ltd, version 2.3.0). To convert the measured optical reflection-interference contrast into a molecular mass, a known protein size marker (NativeMarkTM Unstained Protein Standard, Invitrogen) was measured the same day. All samples were measured in triplicates.

**Magnetic tweezers.** Magnetic tweezers measurements were carried out in a custom-built magnetic tweezers setup at room temperature[56,57]. The DNA constructs were bound by their biotinylated end to 2.8 μm streptavidin-coupled magnetic beads (Dynabeads M280, Thermofischer Scientific) and flushed into the flow cell. Its bottom glass slide was covered with antidigoxigenin to enable the specific attachment of the digoxigenin-modified end of the construct. After a short incubation time, the excess of the magnetic beads was flushed away. Lowering the magnets allowed to stretch DNA molecules tethered to the magnetic beads. The measurements were performed at 30 Hz using video microscopy and real-time GPU accelerated image analysis[57]. Magnetic forces were calibrated using fluctuation analysis[58]. DNA unwinding experiments were performed in reaction buffer (25 mM Tris-acetate pH 7.5, 2 mM magnetic acetate, 50 mM KCl, 1 mM ATP, 1 mM DTT, 0.1 mg/ml BSA) containing 5 nM nuclease-dead Dna2 E675A and 20 nM wild type RPA or fragments. Analysis was performed using a custom-written MATLAB program[40] (see also https://doi.org/10.5281/zenodo.5524562). Briefly, traces acquired in the magnetic tweezers measurements were divided into fragments of constant velocity. Each velocity was calculated from the linear fit of such fragments. Mean velocity and standard error were calculated from all the velocities of the fragments.

**Reporting summary.** Further information on research design is available in the Nature Research Reporting Summary linked to this article.

## Data availability

The data that support this study are available from the corresponding author upon reasonable request. Relevant data generated or analyzed during this study are included in this article and its supplementary information. Source data are provided with this paper, and at Zenodo: https://doi.org/10.5281/zenodo.5524562. The structural models data were deposited in the ModelArchive database and are available: https://modelarchive.org/doi/10.5452/ma-q8w8e. Source data are provided with this paper.

## Code availability

The custom-made Matlab code for the analysis of magnetic tweezers data is available at Zenodo: https://doi.org/10.5281/zenodo.5524562.

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

## Acknowledgements

We thank members of the Cejka lab for critical comments, Marc Wold (University of Iowa) for the yeast RPA expression construct and Luca Varani (Institute for Research in Biomedicine, Bellinzona) for the pET21 construct. This work was supported by the Swiss National Science Foundation (SNSF) Grant 31003A_175444, European Research Council (ERC) Grant 681630 to P.C, ERC Grant 724863 to R.S. and by the French Infrastructure for Integrated Structural Biology (FRISBI) ANR-10-INSB-05-01; CHIPSET ANR-15-CE11-0008-01 to R.G.

## Author contributions

A.A. and P.C. designed, and A.A. performed all biochemical and yeast experiments. K.K. and R.S. designed and performed magnetic tweezer experiments. M.G. and R.S. designed and performed mass photometry experiments. V.K. designed the SSL mutant. R.G. performed the molecular docking. A.A. and P.C. wrote the manuscript. All authors contributed to prepare the final version of the manuscript.

## Competing interests

The authors declare no competing interests.
