## [Peer Review File · Nature Communications]

Distinct RPA domains promote recruitment and the helicase-nuclease activities of Dna2Reviewers' Comments:

Reviewer #1:

Remarks to the Author:

Here, Acharya et al investigate the mechanisms by which RPA stimulates resection mediated by Dna2. They present bulk and single molecule biochemical data showing that the ability of RPA to stimulate Dna2's nuclease and helicase activities is separable from its ability to recruit Dna2 to DNA. They identify a region of RPA in the N-terminus of Rfa1 in which mutations impair nuclease activity stimulation but not recruitment of Dna2. The authors also find that RPA's ability to bind DNA is tightly coupled to blocking of Dna2's 3' flap nuclease activity, suggesting that RPA oriented "backwards" on the DNA is a nonspecific block to Dna2's nuclease activity much like mtSSB, used as a control in these experiments. A mutant that is deficient in stimulation of Dna2's helicase activity, Rfa1-SKYD, is also identified.

The manuscript reports elegant mechanistic experiments. The results are impeccable and help answer open questions regarding regulation of Dna2 function by RPA. With proper revisions, the study will be of considerable interest to the DNA repair/replication community.

Specific points

1. The authors could test key mutants (i.e. Rfa1-SSL, SKYD, etc.) in yeast resection assays. This would shed light on how important recruitment vs. stimulation of Dna2 is within the context of DNA end resection in cells.
2. In Figure 1B, RPA and mtSSB should be tested with the same concentration of Dna2 and the resulting data plotted together in one graph.
3. It is unclear why an alignment of Dna2 is shown in Figure 4C. This alignment is not really discussed in the paper. An alignment of Rfa1 showing the SSL region might be more informative here.
4. To rule out other effects on resection, key RPA mutants (e.g., SSL and SKYD) should be tested for interaction with Dna2 using pulldown assays, as well as their ability to stimulate DNA unwinding by Sgs1.
5. In reference to Fig. 2a, the authors claim "RPA facilitates the binding of Dna2 to DNA." However, the experiments in Fig. 2a do not distinguish between Dna2 bound directly to DNA and Dna2 bound to RPA without making contact with the DNA directly. Do the authors have evidence that Dna2 and RPA occupy the same strand of DNA as the schematic in Fig. 2a suggests?
6. The authors claim that Rfa1-C is inhibitory, but Rfa1-ABC and Rfa1-AB have similar levels of helicase activity in Fig. 5k. If Rfa1-F or the combination of Rfa1-F and Rfa1-C inhibits helicase activity, how do the authors explain this phenomenon in the context of full-length RPA?
7. The representative molecules in Fig. 2c should match the images used in previous figures or defined in the legend.

Reviewer #2:

Remarks to the Author:

In this manuscript, Acharya and co-workers utilize a combination of biochemistry, single molecule biophysics and molecular modeling to explore the interactions between the single-strand DNA binding protein RPA (replication protein A) and the helicase/nuclease protein Dna2. The result of their work provides a clear demonstration that RPA is not simply a passive player in the reactions catalyzed by Dna2 but is instead a complex regulatory factor that can affect the properties of Dna2 through multiple distinct mechanisms. Indeed, this work has revealed a rich network of functional interactions between RPA and Dna2, and immediately raises the question of whether similar types of RPA-mediated regulatory effects might be found for other reactions in which RPA is a necessary participant. The paper is well written and should be very well received by many scientists working on various aspects of DNA metabolism. It will make an important addition to the literature.

Below, I have some minor comments and suggestions that could be addressed prior to publication.

Page 3, paragraph 1: In the initial discussion of RPA within the introduction, it might be useful for some readers if there were a simple schematic in Figure 1 that the authors could refer to when describing the various protein domains.

Page 3, paragraph 3: "Dna2 may have an additional lesser-defined function..."

Page 4, paragraph 2: Awkward sentence: "We show here that RPA promotes Dna2 on top of its recruitment function." – perhaps rewrite this sentence?

Page 11, last sentence: "This technique allows one to estimate..."

Page 12, paragraph 1: "... we observed only mass peaks corresponding to single RPA and Dna2 molecules (Fig. 2c)" → Should this be Fig. 2d?

Page 12, paragraph 2 / Fig. 2f: Can the authors clarify whether RPA or mtSSB were added at the exact same time as the ATP?

Page 13, paragraph 2: Just to be sure, the Rfa1-F domain alone was soluble and monomeric, correct? Perhaps best to specify this point in the text.

Page 15 & other: For experiments Dna2-E675A, have the author considered testing whether the presence of RPA alters its ATP hydrolysis activity? If RPA enhanced ATP turnover by Dna2, then it might provide a nice explanation for many of the authors' observations. This type of assay could also be useful in the analysis of Rfa1-FAB and some of the other mutants.

Page 16, first sentence: I will leave this issue to the authors' discretion, but in my view the sentence that "RPA promotes unwinding... in a species-specific manner..." is too strong of a statement. mtSSB is not simply another form of RPA, instead it is really a completely different protein. Had the comparison been between human and yeast RPA, then the statement would be fine, but as is I would suggest erring on the side of caution.

Figure 1b: Can the authors clarify why the concentration of Dna2 was increased from 30 to 45 pM in reactions with RPA and mtSSB, respectively. This information can be put in the methods.

Figure 5f, I & Supplementary Figure 5e, h : Can the authors' clarify how the error was calculated for the velocity distributions. The +/- 3 bp/sec value reported for all of the data sets seems like it may not be correct (the distributions seem to be much broader than would be consistent with the small error, but maybe I am missing some nuance of the analysis).

Figure 6d, e : Is there quantitation corresponding to these gels?

Figure 7c: Shouldn't the RPA complexes bound to the ssDNA be shown in the opposite orientations for the 5' and 3' strands to indicate binding polarity?

Reviewer #3:

Remarks to the Author:

Acharya et. al, using detailed biochemical and single molecule approaches, show the complex interplay between RPA and Dna2 in regulating the nuclease and helicase activities of Dna2. Elegant structural studies from the Pavletich group had shown the interactions between RPA and Dna2 promote the

proper alignment of Dna2 on DNA such that the polarity of RPA dictated the proper positioning of Dna2 for appropriate nucleolytic cleavage. Nevertheless, whether RPA solely functioned to recruit Dna2 onto DNA and how RPA regulated Dna2 activity were puzzling. In this study, the authors show that RPA possesses two distinguishable roles: one to regulate the binding and recruitment of Dna2 and the other to control its nuclease and helicase activities. They map these activities to different regions of RPA and using specific mutations are able to selectively impair these activities.

This is detailed work which are carried out well. The work is complex and the authors have done an admirable job of laying out the results as clearly as possible. My comments and suggestions are below. My primary concern resides in the use of the individual Rfa1 subunits which are known to be unstable in solution when isolated. If the authors can for certain show that these polypeptides are well behaved single-species in solution, then I would be willing to accept their conclusions.

Introduction, first paragraph: 'The key RPA functions are to..'

Introduction, first paragraph: cofactor (non hyphenated).

Introduction, last paragraph: This line is missing text, a verb. "We show here that RPA promotes Dna2 on top of its recruitment function".

Results, 2nd paragraph: Misreferred Fig. "when RPA was added to Dna2 in the absence of DNA, we observe mass peaks....(Fig. 2d)" – not 2c.

Second paragraph in results under "The Rfa1-FAB domains stimulate the Dna2 nuclease" section, there is a statement: "These experiments demonstrated that the capacity of RPA to bind ssDNA does not directly correspond to the ability to stimulate the Dna2 nuclease". This statement is not entirely accurate. While the mutations in Rfa1-C reduce the affinity of RPA, we are still talking about DNA binding affinity in the 2-4 nM range. The observed differences are likely due to alterations in the physical interactions, but the role for DNA interactions cannot be ruled out.

Supplementary Figure 3 has some weird data:

1. Rfa1 runs at around 66-75 kDa in the gels shown (S3b and S3g). The isolated bands in S3i and S3j do not match up in sizes. Are the ladders labeled incorrectly?
2. Another perplexing set of data are in S3k. Rfa1 binds to DNA (good), Rfa2 does not bind to DNA (?). Rfa3 binds to DNA (????). Or, are the gel images flipped?
3. The data using the isolated Rfa1, Rfa2, and Rfa3 proteins are interesting. In our hands and in that of the Wold laboratory, the large subunit does not behave well when isolated separately (precipitates, etc). Similarly, 32 and 14 have to be coexpressed as a complex for well behaved protein. The fact that the authors have been able to get these polypeptides to be soluble with just a his-tag is certainly perplexing. And the fact that Rfa2 does not bind to DNA points to these issues.
4. It might just be better if the authors used only the FAB dataset to prove their point. Clearly, this fragment is sufficient to stimulate the nuclease activity of Dna2. I am uncomfortable with the isolated subunits having knowledge about how they poorly behave in solution.

The section discussing the results of the unwinding experiments are also a bit perplexing. Just to point out, I am not disagreeing with what the authors are showing nor do I have a better explanation. I just want to point this out. The FABC does not promote unwinding, the ABC promotes unwinding, and the full RPA complex promotes unwinding. FAB, ABC, and AB promotes unwinding. Does this mean that there are interactions between F and C that together (and as shown, alone) block unwinding? Or is the FABC data an artifact because without Rfa2 and 3, a large hydrophobic patch would be open and likely causing a non-specific interaction with F which then in turn is inhibiting the unwinding activity. Can the authors somehow show that the FABC, Rfa2 and Rfa3 proteins are properly folded?

Does the AB fragment directly interact with Dna2? Can this be experimentally tested?

Can the authors show that the isolated Rfa1, Rfa2, and Rfa3 proteins behave as single species in solution using their mass photometer? Also, can they do the same comparison for the FABC and the ABC versions?

Response to reviewers

"Distinct RPA domains promote recruitment and the helicase-nuclease activities of Dna2"

We would like to thank the reviewers for their time and interest in reading our manuscript, as well as for providing very insightful and constructive inputs. Please find below how we addressed the individual points.

Reviewer #1 (Remarks to the Author):

Here, Acharya et al investigate the mechanisms by which RPA stimulates resection mediated by Dna2. They present bulk and single molecule biochemical data showing that the ability of RPA to stimulate Dna2's nuclease and helicase activities is separable from its ability to recruit Dna2 to DNA. They identify a region of RPA in the N-terminus of Rfa1 in which mutations impair nuclease activity stimulation but not recruitment of Dna2. The authors also find that RPA's ability to bind DNA is tightly coupled to blocking of Dna2's 3' flap nuclease activity, suggesting that RPA oriented "backwards" on the DNA is a nonspecific block to Dna2's nuclease activity much like mtSSB, used as a control in these experiments. A mutant that is deficient in stimulation of Dna2's helicase activity, Rfa1-SKYD, is also identified.

The manuscript reports elegant mechanistic experiments. The results are impeccable and help answer open questions regarding regulation of Dna2 function by RPA. With proper revisions, the study will be of considerable interest to the DNA repair/replication community.

> Specific points

> 1. The authors could test key mutants (i.e. Rfa1-SSL, SKYD, etc.) in yeast resection assays. This would shed light on how important recruitment vs. stimulation of Dna2 is within the context of DNA end resection in cells.

Answer: In order to test the key RPA mutants in yeast resection assays, we set out to create point mutants in the *S. cerevisiae* JKM139 strain in the *exo1Δ* background to monitor at Dna2-mediated resection. We used standard procedures for CRISPR-Cas9 based genetic manipulations. However, we failed to obtain both RPA-SSL and RPA-SKYD mutants in this background, suggesting that the mutations bring about a lethal phenotype.

We first sequenced 14 and 13 clones for the SSL and SKYD mutants, respectively, in the JKM139 *exo1Δ* cells. In all cases, we found the PAM motif mutated, but the target residues were unchanged. To rule out any effects related to the lack of Exo1, we next proceeded with the wild type JKM139 background. We tested again 14 and 13 clones as above, but the result was the same. We also tried to create the same mutants in the JKM139 *pif1-m2* background (the *pif1-m2* mutation is a known suppressor of *dna2Δ* lethality), with the expectation that the Dna2 dependent effects of these mutants could be potentially counteracted by the *pif1-m2* mutation. Again, we failed to obtain the RPA-SSL mutant in this background despite testing 12 clones. In order to exclude any technical reasons for the failure to obtain the SSL mutant, we created silent mutations of the same amino acids (11, 14, and 15) using the same approach, and obtained silently mutated alleles in all of the 7 clones tested. Therefore, the SSL mutation is most

likely lethal, and affects RPA function beyond the interplay with Dna2. We would like to point out that the I14S mutation, which is one of the three point mutations in the SSL variant, already leads to a greatly reduced growth rate, despite still retaining viability (Umezu et al., 1998).

Regarding the RPA-SKYD mutation, we could obtain two clones in the JKM139 *pif1-m2* background out of 10 clones tested. However, combination of this mutation with *exo1Δ* resulted in lethality. We also noted that the two clones we obtained differed dramatically in growth rates, suggesting that the mutants might be acquiring suppressor mutations, making the analysis unreliable. Therefore, we were not able to perform the resection assays. We note our unsuccessful attempts to create these mutants in the manuscript: "We note that we failed to construct the RPA-SSL and RPA-SKYD mutants in yeast cells, suggesting a likely lethal phenotype, in agreement with already notably reduced growth rate of RPA-I14S (Umezu et al., 1998)". We agree that these experiments would have been informative to fully understand the regulation of Dna2-mediated long-range resection in yeast. However, it is not quite unexpected that the mutations in the essential RPA factor might affect its additional functions. We hope that the reviewer agrees that further analysis would be above the scope of the current manuscript.

> 2. In Figure 1B, RPA and mtSSB should be tested with the same concentration of Dna2 and the resulting data plotted together in one graph.

Answer: The reason we designed the experiment with different Dna2 concentrations was to better visualize stimulation by RPA (using lower Dna2 concentration, leading to a smaller proportion of product) and inhibition with SSB (using higher Dna2 concentration). However, we also show (Figure R1) that the same trend was observed also when the same Dna2 concentrations of Dna2 were used. We note in the manuscript: "the same results were also obtained with other Dna2 concentrations."

Figure R1: Yeast RPA and human mitochondrial SSB (mtSSB) were used in nuclease assays with 45 pM Dna2 and 5'-overhanged DNA substrate (45 nt ssDNA, 48 bp dsDNA, 1 nM, in molecules).

> 3. It is unclear why an alignment of Dna2 is shown in Figure 4C. This alignment is not really discussed in the paper. An alignment of Rfa1 showing the SSL region might be more informative here.

Answer: We agree with the reviewer and have updated the alignment in the Figure 4c.

> 4. To rule out other effects on resection, key RPA mutants (e.g., SSL and SKYD) should be tested for interaction with Dna2 using pulldown assays, as well as their ability to stimulate DNA unwinding by Sgs1.

Answer:

We tested the ability of the key RPA mutants (SSL and SKYD) to stimulate DNA unwinding by Sgs1 and observed that they were comparable. We comment in the text “We also noted that these mutants were comparable in terms of their capacity in stimulating the Sgs1 helicase, suggesting that the defects of these mutants in resection are majorly due to the impaired interplay with the Dna2 helicase-nuclease (Supplementary Fig. 7a, b)”.

Regarding the physical interactions of the RPA mutants with Dna2: The RPA-SSL mutant consists of three mutations: F11S, I14S, and F15L. It was previously shown that the I14S mutation in yeast Rfa1 displays reduced interaction with yeast Dna2 (Bae *et al.*, NAR, 2003). So, we expect that the SSL mutant is also impaired in physical interaction with Dna2 since the additional mutations further disrupt the conserved helix around the I14 residue. RPA-SKYD is mutated in only in the secondary interaction sites of RPA-Dna2 pair and hence, we would anticipate that RPA-SKYD might be partly, if not fully proficient in Dna2 interaction since its major Dna2 interacting region in the N terminus is intact. To circumvent this point, we tried to test the interaction between yeast Dna2 and Rfa1-AB/Rfa1-AB-SKYD (which lacks the primary interaction interface), but failed to capture the interaction in a western blot-based pulldown assay even with the intact Rfa1-AB domain. We note that physical interaction between mouse RPA-AB fragment and mouse Dna2 OB domain (residues 21 to 122) was reported using isothermal titration calorimetry assays (Zhou *et al.*, eLife, 2015), albeit with very high K_d . We believe that the likely very transient interaction is lost during the washing steps of our pulldown assay.

> 5. *In reference to Fig. 2a, the authors claim “RPA facilitates the binding of Dna2 to DNA.” However, the experiments in Fig. 2a do not distinguish between Dna2 bound directly to DNA and Dna2 bound to RPA without making contact with the DNA directly. Do the authors have evidence that Dna2 and RPA occupy the same strand of DNA as the schematic in Fig. 2a suggests?*

Answer: We observed that at the same concentration of Dna2, Dna2 is recruited to DNA only when the ssDNA overhang was pre-coated with RPA and not to naked DNA (Figure 2a). In addition, we performed mass photometry assays to show that RPA remains present as a part of the nucleoprotein (Dna2-DNA-RPA) ternary complex (which was controversial to-date). We presume that Dna2 first forms contacts with RPA - we used saturating concentrations of RPA and the ssDNA segment is fully coated with RPA. Whether Dna2 then remains associated with RPA, or repositions on the substrate, we cannot distinguish in our assays. Under nuclease-permissive conditions, Dna2 would degrade the overhanged DNA starting from the 5' ssDNA end, so we would not expect a major repositioning of the proteins. What is the exact structure of the ternary complex would be an interesting topic to investigate.

> 6. *The authors claim that Rfa1-C is inhibitory, but Rfa1-ABC and Rfa1-AB have similar levels of helicase activity in Fig. 5k. If Rfa1-F or the combination of Rfa1-F and Rfa1-C inhibits helicase activity, how do the authors explain this phenomenon in the context of full-length RPA?*

Answer: Our observation that Rfa1-C or the combination of Rfa1-F and Rfa1-C inhibit helicase activity is based on the direct comparison of the fragments of Rfa1 in bulk and single molecule assays. In the context of full-length RPA, Rfa1-C forms a trimeric core with Rfa2 and Rfa3. Therefore, the conformation of the protein/protein complex and exposed residues will differ between Rfa1 (in particular Rfa1-C, and Rfa1

truncations) and RPA. While these observations may not be directly physiologically relevant, we have used them as a guide to identify the key patches involved in the specific interplay between RPA and Dna2 helicase/nuclease. The point mutants were then validated in the context of the full-length trimeric RPA.

When discussing the apparent inhibitory effects of Rfa1-F and Rfa1-C on DNA unwinding, we added a note that "we cannot exclude that some of these effects may be caused by instability of the Rfa1 fragments".

> 7. *The representative molecules in Fig. 2c should match the images used in previous figures or defined in the legend.*

Answer: Thank you. This is included and updated.

Reviewer #2 (Remarks to the Author):

In this manuscript, Acharya and co-workers utilize a combination of biochemistry, single molecule biophysics and molecular modeling to explore the interactions between the single-strand DNA binding protein RPA (replication protein A) and the helicase/nuclease protein Dna2. The result of their work provides a clear demonstration that RPA is not simply a passive player in the reactions catalyzed by Dna2 but is instead a complex regulatory factor that can affect the properties of Dna2 through multiple distinct mechanisms. Indeed, this work has revealed a rich network of functional interactions between RPA and Dna2, and immediately raises the question of whether similar types of RPA-mediated regulatory effects might be found for other reactions in which RPA is a necessary participant. The paper is well written and should be very well received by many scientists working on various aspects of DNA metabolism. It will make an important addition to the literature.

Below, I have some minor comments and suggestions that could be addressed prior to publication.

> *Page 3, paragraph 1: In the initial discussion of RPA within the introduction, it might be useful for some readers if there were a simple schematic in Figure 1 that the authors could refer to when describing the various protein domains.*

Answer: We now included the schematic into Figure 1.

> *Page 3, paragraph 3: "Dna2 may have an additional lesser-defined function..."*

Answer: Thank you. This is now included and updated.

> *Page 4, paragraph 2: Awkward sentence: "We show here that RPA promotes Dna2 on top of its recruitment function." – perhaps rewrite this sentence?*

Answer: This is rephrased and updated as "We show here that RPA promotes the catalytic activities of Dna2 in addition to its recruitment function".

> *Page 11, last sentence: "This technique allows one to estimate..."*

Answer: Thank you.

> Page 12, paragraph 1: "... we observed only mass peaks corresponding to single RPA and Dna2 molecules (Fig. 2c)" — Should this be Fig. 2d?

Answer: Thank you for spotting this error.

> Page 12, paragraph 2 / Fig. 2f: Can the authors clarify whether RPA or mtSSB were added at the exact same time as the ATP?

Answer: They were added at the same time. This is now clarified in the text.

> Page 13, paragraph 2: Just to be sure, the Rfa1-F domain alone was soluble and monomeric, correct? Perhaps best to specify this point in the text.

Answer: Please see our detailed response to reviewer #3, who raised similar concerns. During the preparation of the Rfa1 fragments, we sometimes observed aggregation and precipitation, however the insoluble fraction was later removed from the sample. The fragments used in our analysis, including Rfa1-F, were soluble and exhibited low aggregation index (based on absorbance at 280 vs 350). We note that the fragments were used to identify the key interaction sites, and the key mutants were subsequently created in heterotrimeric RPA that behaved well.

> Page 15 & other: For experiments Dna2-E675A, have the author considered testing whether the presence of RPA alters its ATP hydrolysis activity? If RPA enhanced ATP turnover by Dna2, then it might provide a nice explanation for many of the authors' observations. This type of assay could also be useful in the analysis of Rfa1-FAB and some of the other mutants.

Answer: Thank you. We tested for the effect of the RPA variants (SSL and SKYD) on the ATP hydrolytic activity of Dna2-E675A (nuclease-dead variant). We observed that RPA and RPA-SSL were competent in stimulating the ATPase activity of Dna2-E675A, whereas RPA-SKYD was defective in this regard. We performed the experiments with full length RPA variants because they are more physiologically relevant. These data are included in Fig. 6i. We include *"Following the defects of RPA-SKYD in promoting the helicase activity of Dna2-E675A, we wanted to characterize if RPA has similar effects on the capacity of Dna2-E675A to hydrolyze ATP. To this point, we used a spectrophotometric assay, which measures ATPase activity based on a reaction coupled to the oxidation of NADH. We compared the ATPase activity of Dna2-E675A in the presence of the various RPA point mutants. We observed that the apparent k_{cat} , which is the apparent ATP turnover number, for Dna2-E675A was stimulated approximately 2-fold in the presence of RPA wild type and RPA-SSL. However, the RPA-SKYD mutant did not stimulate the ATPase of Dna2-E675A"*.

> Page 16, first sentence: I will leave this issue to the authors' discretion, but in my view the sentence that "RPA promotes unwinding... in a species-specific manner..." is too strong of a statement. mtSSB is not simply another form of RPA, instead it is really a completely different protein. Had the comparison been between human and yeast RPA, then the statement would be fine, but as is I would suggest erring on the side of caution.

Answer: We agree with the reviewer. Previously, yeast and human RPA were compared with regard to the stimulation of the nuclease activity of Dna2, and the cognate factors always promoted DNA

degradation better. The same is true when looking at the helicase function (Figure R2). So, there is certainly a species-specific component, but non-cognate RPA remains better than SSB. Therefore, we replaced "species-specific" with "specific". As the manuscript is already quite dense, we opted not to include these data.

Figure R2: Yeast and human RPA were used in helicase assays with 5 nM Dna2-E675A and 2.2 kb double-stranded DNA substrate (0.1 nM, in molecules).

Figure 1b: Can the authors clarify why the concentration of Dna2 was increased from 30 to 45 pM in reactions with RPA and mtSSB, respectively. This information can be put in the methods.

Answer: We used low concentrations of Dna2 to obtain less initial degradation of DNA to be able to better see stimulation in the presence of RPA. Similarly, we used high concentrations of Dna2 to obtain more initial degradation of DNA to be able to see inhibition better in the presence of mtSSB (see also **Figure R1**, the same effects were observed with an identical Dna2 concentration).

> Figure 5f, l & Supplementary Figure 5e, h: Can the authors' clarify how the error was calculated for the velocity distributions. The +/- 3 bp/sec value reported for all of the data sets seems like it may not be correct (the distributions seem to be much broader than would be consistent with the small error, but maybe I am missing some nuance of the analysis).

Answer: We now describe the analysis better in the methods. "Traces acquired in the magnetic tweezers measurements were divided into fragments of constant velocity. Each velocity was calculated from the linear fit of such fragments. Mean velocity and standard error were calculated from all the velocities of the fragments". As standard error is standard deviation divided by square root of the number of measurements, we note that the standard deviation would be much broader as pointed out by the reviewer. Yet, the given value is the standard error and hence, it is approximately 10-fold narrower than standard deviation, because we have on the order of 100 measurements for each construct.

> Figure 6d, e: Is there quantitation corresponding to these gels?

Answer: We show one out of three independent experiments in Figures 6d, e. The percentage of unwinding was low, yet consistent. As the figures and related supplementary figures already contained many panels, we decided not to include these quantitations. This decision is also based on our effort to focus the analysis on the full-length RPA variants (where all quantitations are presented). We included the following statement in the legends: "The experiment was performed three times with similar results". We also moved the e panel into supplement, used show instead the nice ATPase result with the heterotrimeric RPA point mutants.

> Figure 7c: Shouldn't the RPA complexes bound to the ssDNA be shown in the opposite orientations for the 5' and 3' strands to indicate binding polarity?

Answer: Thank you, of course. Done.

Reviewer #3 (Remarks to the Author):

> Acharya et. al, using detailed biochemical and single molecule approaches, show the complex interplay between RPA and Dna2 in regulating the nuclease and helicase activities of Dna2. Elegant structural studies from the Pavletich group had shown the interactions between RPA and Dna2 promote the proper alignment of Dna2 on DNA such that the polarity of RPA dictated the proper positioning of Dna2 for appropriate nucleolytic cleavage. Nevertheless, whether RPA solely functioned to recruit Dna2 onto DNA and how RPA regulated Dna2 activity were puzzling. In this study, the authors show that RPA possesses two distinguishable roles: one to regulate the binding and recruitment of Dna2 and the other to control its nuclease and helicase activities. They map these activities to different regions of RPA and using specific mutations are able to selectively impair these activities.

> This is detailed work which are carried out well. The work is complex and the authors have done an admirable job of laying out the results as clearly as possible. My comments and suggestions are below. My primary concern resides in the use of the individual Rfa1 subunits which are know to be unstable in solution when isolated. If the authors can for certain show that these polypeptides are well behaved single-species in solution, then I would be willing to accept their conclusions.

> Introduction, first paragraph: "The key RPA functions are to.."

Answer: This is corrected and updated.

> Introduction, first paragraph: cofactor (non hyphenated).

Answer: Done.

> Introduction, last paragraph: This line is missing text, a verb. "We show here that RPA promotes Dna2 on top of its recruitment function".

Answer: Done.

> Results, 2nd paragraph: Misreferred Fig. "when RPA was added to Dna2 in the absence of DNA, we observe mass peaks...(Fig. 2d)" – not 2c.

Answer: Done.

> Second paragraph in results under “The Rfa1-FAB domains stimulate the Dna2 nuclease” section, there is a statement: “These experiments demonstrated that the capacity of RPA to bind ssDNA does not directly correspond to the ability to stimulate the Dna2 nuclease”. This statement is not entirely accurate. While the mutations in Rfa1-C reduce the affinity of RPA, we are still talking about DNA binding affinity in the 2-4 nM range. The observed differences are likely due to alterations in the physical interactions, but the role for DNA interactions cannot be ruled out.

Answer: We agree and therefore removed this sentence. The next sentence concludes that the stimulation of the Dna2 nuclease requires physical interactions.

> Supplementary Figure 3 has some weird data:

> 1. Rfa1 runs at around 66-75 kda in the gels shown (S3b and S3g). The isolated bands in S3i and S3j do not match up in sizes. Are the ladders labeled incorrectly?

Answer: The reviewer is correct that the label of the ladder was moved and positioned incorrectly. This was corrected and updated. Thank you.

> 2. Another perplexing set of data are in S3k. Rfa1 binds to DNA (good), Rfa2 does not binds to DNA (?). Rfa3 binds to DNA (????). Or, are the gel images flipped?

Answer: No, the gels were not flipped. The binding of both Rfa2 and Rfa3 subunits at the lower protein concentrations was low, much lower compared to Rfa1.

When higher protein concentrations were used for Rfa2 (**Figure R3**), we could detect DNA binding. The amount of Rfa3 we were able to obtain was very low, which did not allow us to perform experiments with higher concentrations.

> 3. The data using the isolated Rfa1, Rfa2, and Rfa3 proteins are interesting. In our hands and in that of the Wold laboratory, the large subunit does not behave well when isolated separately (precipitates, etc). Similarly, 32 and 14 have to be coexpressed as a complex for well behave protein. The fact that the authors have been able to get these polypeptides to be soluble with just a his-tag is certainly perplexing. And the fact that Rfa2 does not bind to DNA points to these issues.

Answer: We agree with the reviewer that the purification of the RPA subunits is not trivial. We also observed precipitation during the preparation, but, after optimizations, we obtained usable amounts of soluble protein for all the three subunits and the respective fragments (although the yield was much lower than for the heterotrimeric RPA). We note that all fragments were purified by his-tag based affinity chromatography followed by anion exchange chromatography. Only the Rfa1 subunit (and the corresponding point mutants in Rfa1) was extensively washed and purified with one step (his-tag), to obtain higher yield. We saw identical specific activities of the Rfa1 protein purified by the two different methods.

> 4. It might just be better if the authors used only the FAB dataset to prove their point. Clearly, this fragment is sufficient to stimulate the nuclease activity of Dna2. I am uncomfortable with the isolated subunits having knowledge about how they poorly behave in solution.

Answer: We understand the concerns of the reviewer. We would like to point out that all the experiments with the fragments were performed with the intent to obtain preliminary understanding of the requirement for the individual domains. Later, we build on these initial findings to define the function of the full-length trimeric RPA point mutants, which behave very well. However, we note that each of our key Rfa1 fragments (Rfa1-FABC, Rfa1-FAB, Rfa1-AB, Rfa1ABC) was active in at least one assay (stimulation of the Dna2 nuclease or helicase, or Dna2 recruitment), beyond ssDNA binding. So, although there might be some caveats, we believe we have sufficient evidence to exclude that the fragments are completely misfolded/inactive.

> The section discussing the results of the unwinding experiments are also a bit perplexing. Just to point out, I am not disagreeing with what the authors are showing nor do I have a better explanation. I just want to point this out. The FABC does not promote unwinding, the ABC promotes unwinding, and the full RPA complex promotes unwinding. FAB, ABC, and AB promotes unwinding. Does this mean that there are interactions between F and C that together (and as shown, alone) block unwinding? Or is the FABC data an artifact because without Rfa2 and 3, a large hydrophobic patch would be open and likely causing a non-specific interaction with F which then in turn is inhibiting the unwinding activity. Can the authors somehow show that the FABC, Rfa2 and Rfa3 proteins are properly folded?

Answer: We agree with the reviewer that the interpretation of these data is not trivial. As noted below, in mass photometry, Rfa1 is largely monomeric (as well as Rfa1-FAB and Rfa1-ABC). Also, in the nuclease assays, Rfa1 promoted Dna2 equally well as the full-length RPA. As Rfa1-F is required to promote the nuclease, we would assume that this domain will also be properly folded within Rfa1. As far as the apparent inhibition by the C subunit, please see our answer to Reviewer #1. Indeed, it is possible that when the C subunit is not in complex with Rfa2 and Rfa3, it may behave in a different and non-physiological way in contrast to when it is within full-length RPA.

When discussing the apparent inhibitory effects of Rfa1-F and Rfa1-C on DNA unwinding, we added a note that "we cannot exclude that some of these effects may be caused by instability of the Rfa1 fragments".

However, we again point out that the fragment analysis was used as a stepping stone for the design of the point mutants within the RPA trimer (please see also our reasoning above).

> Does the AB fragment directly interact with Dna2? Can this be experimentally tested?

Answer: We tested the AB fragment for direct interaction with Dna2. However, we could not observe any detectable binding in pulldown assays under our conditions, indicating that the previously reported interaction (Zhou *et al.*, eLife, 2015) is likely very transient. Please see our answer to reviewer number 1 for more details.

> Can the authors show that the isolated Rfa1, Rfa2, and Rfa3 proteins behave as single species in solution using their mass photometer? Also, can they do the same comparison for the FABC and the ABC versions?

Answer: We performed mass photometric analysis of the Rfa1 (i.e., Rfa1-FABC), Rfa1-ABC and Rfa1-FAB fragments, which are most relevant for our analysis. We observed that these proteins are largely monomeric in solution (**Figure R4**). Mass photometric analysis could not be performed for the Rfa2 and Rfa3 subunits, because their molecular weight is below the minimum required by the instrument.

Figure R4: Measured molecular weight distributions of Rfa1, Rfa1-ABC and Rfa1-FAB complexes.

Reviewers' Comments:

Reviewer #1:

Remarks to the Author:

Authors have made sincere efforts to address the few issues raised, including suggestions for new experiments. Alas, not every new experiment has worked because of cell viability and other issues, but this is not the authors' fault though.

The biochemical analysis is impeccable and the model proposed, even without genetics, is plausible and novel.

The Cejka lab has made great contributions to help solve riddles regarding how DNA end resection works during homologous recombination and DNA break repair in eukaryotes, and this is yet another splendid study that deserves to be published in Nature Communications.

Reviewer #2:

Remarks to the Author:

The authors have fully addressed all of my comments. This is an outstanding study and will make an important contribution to the literature.

Reviewer #3:

Remarks to the Author:

My concerns have been clarified.